# Wnt signaling mediates acquisition of blood–brain barrier properties in naïve endothelium derived from human pluripotent stem cells

Benjamin D Gastfriend[1], Hideaki Nishihara[2], Scott G Canfield[1†], Koji L Foreman[1], Britta Engelhardt[2], Sean P Palecek[1]*, Eric V Shusta[1,3]*

[1]Department of Chemical and Biological Engineering, University of Wisconsin–Madison, Madison, United States; [2]Theodor Kocher Institute, University of Bern, Bern, Switzerland; [3]Department of Neurological Surgery, University of Wisconsin–Madison, Madison, United States

*For correspondence:
sppalecek@wisc.edu (SPP);
eshusta@wisc.edu (EVS)

Present address: †Department of Anatomy, Cell Biology, and Physiology, Indiana University School of Medicine, Terre Haute, United States

**Abstract** Endothelial cells (ECs) in the central nervous system (CNS) acquire their specialized blood–brain barrier (BBB) properties in response to extrinsic signals, with Wnt/β-catenin signaling coordinating multiple aspects of this process. Our knowledge of CNS EC development has been advanced largely by animal models, and human pluripotent stem cells (hPSCs) offer the opportunity to examine BBB development in an in vitro human system. Here, we show that activation of Wnt signaling in hPSC-derived naïve endothelial progenitors, but not in matured ECs, leads to robust acquisition of canonical BBB phenotypes including expression of GLUT-1, increased claudin-5, decreased PLVAP, and decreased permeability. RNA-seq revealed a transcriptome profile resembling ECs with CNS-like characteristics, including Wnt-upregulated expression of *LEF1*, *APCDD1*, and *ZIC3*. Together, our work defines effects of Wnt activation in naïve ECs and establishes an improved hPSC-based model for interrogation of CNS barriergenesis.

## Editor's evaluation

This work provides a considerable novelty to the view on barrier induction in endothelial cells in the central nervous system by the Wnt/β-catenin pathway. By recapitulating the endothelial differentiations events towards a blood–brain barrier phenotype, the presented results offer novel ways to establish cell culture systems that faithfully recapitulate the blood–brain barrier ex vivo, and thereby supporting basic as well as translational research.

## Introduction

In the central nervous system (CNS), vascular endothelial cells (ECs) are highly specialized, with complex tight junctions, expression of a spectrum of nutrient and efflux transporters, low rates of vesicle trafficking, no fenestrae, and low expression of immune cell adhesion molecules (*Reese and Karnovsky, 1967*; *Obermeier et al., 2013*). ECs bearing these attributes, often referred to as the blood–brain barrier (BBB), work in concert with the other brain barriers to facilitate the tight regulation of the CNS microenvironment required for proper neuronal function (*Daneman and Engelhardt, 2017*; *Profaci et al., 2020*). During development, the Wnt/β-catenin signaling pathway drives both CNS angiogenesis, during which vascular sprouts originating from the perineural vascular plexus invade the developing neural tube, and the coupled process of barriergenesis by which resulting ECs

**eLife digest** The cells that line the inside of blood vessels are called endothelial cells. In the blood vessels of the brain, these cells form a structure called the 'blood-brain barrier', which allows nutrients to pass from the blood into the brain, while at the same time preventing harmful substances like toxins from crossing. Faults in the blood-brain barrier can contribute to neurological diseases, but the blood-brain barrier can also restrict drugs from accessing the brain, making it difficult to treat certain conditions. Understanding how the endothelial cells that form the blood-brain barrier develop may offer insight into new treatments for neurological diseases.

During the development of the embryo, endothelial cells develop from stem cells. They can also be generated in the laboratory from human pluripotent stem cells or 'hPSCs', which are cells that can produce more cells like themselves, or differentiate into any cell type in the body. Scientists can treat hPSCs with specific molecules to make them differentiate into endothelial cells, or to modify their properties. This allows researchers to monitor how different types of endothelial cells form.

Endothelial cells at the blood-brain barrier are one of these types. During their development, these cells gain distinct features, including the production of proteins called GLUT-1, claudin-5 and LSR. GLUT-1 transports glucose across endothelial cells' membranes, while claudin-5 and LSR tightly join adjacent cells together, preventing molecules from leaking into the brain through the space between cells. In mouse endothelial cells, a signaling protein called Wnt is responsible for turning on the genes that code for these proteins. But how does Wnt signaling impact human endothelial cells?

Gastfriend et al. probed the effects of Wnt signaling on human endothelial cells grown in the lab as they differentiate from hPSCs. They found that human endothelial cells developed distinct blood-brain barrier features when Wnt signaling was activated, producing GLUT-1, claudin-5 and LSR. Gastfriend et al. also found that human endothelial cells were more responsive to Wnt signaling earlier in their development. Additionally, they identified the genes that became activated in human endothelial cells when Wnt signaling was triggered.

These findings provide insight into the development and features of the endothelial cells that form the human blood-brain barrier. The results are a first step towards a better understanding of how this structure works in humans. This information may also allow researchers to develop new ways to deliver drugs into the brain.

begin to acquire BBB properties (*Liebner et al., 2008*; *Stenman et al., 2008*; *Daneman et al., 2009*; *Engelhardt and Liebner, 2014*; *Umans et al., 2017*). Specifically, neural progenitor-derived Wnt7a and Wnt7b ligands signal through Frizzled receptors and the obligate co-receptors RECK and GPR124 (ADGRA2) on ECs (*Kuhnert et al., 2010*; *Cullen et al., 2011*; *Vanhollebeke et al., 2015*; *Cho et al., 2017*; *Eubelen et al., 2018*; *Vallon et al., 2018*). Other ligands function analogously in other regions of the CNS, including Norrin in the retina and cerebellum (*Ye et al., 2009*; *Wang et al., 2012*) and potentially Wnt3a in the dorsal neural tube (*Daneman et al., 2009*). Furthermore, Wnt/β-catenin signaling is required for maintenance of CNS EC barrier properties in adulthood (*Tran et al., 2016*), with astrocytes as a major source of Wnt7 ligands (*He et al., 2018*; *Vanlandewijck et al., 2018*; *Guérit et al., 2021*).

Molecular hallmarks of Wnt-mediated CNS EC barriergenesis are (i) acquisition of glucose transporter GLUT-1 expression, (ii) loss of plasmalemma vesicle-associated protein (PLVAP), and (iii) upregulation of claudin-5 (*Daneman et al., 2009*; *Kuhnert et al., 2010*; *Cho et al., 2017*; *Umans et al., 2017*; *Wang et al., 2019*). Notably, the Wnt-mediated switch between the 'leaky' EC phenotype (GLUT-1$^-$ PLVAP$^+$ claudin-5$^{low}$) and the barrier EC phenotype (GLUT-1$^+$ PLVAP$^-$ claudin-5$^{high}$) correlates with reduced permeability to molecular tracers (*Wang et al., 2012*; *Cho et al., 2017*) and is conserved in multiple contexts. For instance, medulloblastomas that produce Wnt-inhibitory factors have leaky vessels (*Phoenix et al., 2016*). Moreover, vasculature perfusing circumventricular organs is leaky due to low levels of Wnt signaling (*Benz et al., 2019*; *Wang et al., 2019*). Notably, ectopic activation of Wnt in ECs of circumventricular organs induces GLUT-1 and suppresses PLVAP (*Benz et al., 2019*; *Wang et al., 2019*). However, similar ectopic activation of Wnt in liver and lung ECs produces only very minor barriergenic effects (*Munji et al., 2019*), and Wnt activation in cultured primary mouse brain ECs does not prevent culture-induced loss of barrier-associated gene expression (*Sabbagh and*

*Nathans, 2020*). The reasons for the apparent context-dependent impacts of Wnt activation in ECs remain unclear and motivate systematic examination of this process in a simplified model system. Further, given species differences in brain EC transporter expression (*Uchida et al., 2011*), drug permeability (*Syvänen et al., 2009*), and gene expression (*Song et al., 2020*), this process warrants investigation in human cells to complement mouse in vivo studies.

Prior studies have evaluated the impact of Wnt activation in immortalized human brain ECs and observed only modest effects on barrier phenotype (*Paolinelli et al., 2013*; *Laksitorini et al., 2019*). Combined with the aforementioned deficits observed in primary adult mouse brain ECs that are not rescued by ectopic Wnt activation (*Sabbagh and Nathans, 2020*), one possibility is that mature, adult endothelium is largely refractory to Wnt activation, and that Wnt responsiveness is a property of immature ECs analogous to those in the perineural vascular plexus. Human pluripotent stem cells (hPSCs) offer an in vitro human model system for systematic investigation of molecular mechanisms of BBB phenotype acquisition, especially given their ability to model early stages of endothelial specification and differentiation. However, currently available hPSC-based models of CNS endothelial-like cells are not well suited for modeling the BBB developmental progression as they do not follow a developmentally relevant differentiation trajectory, lack definitive endothelial identity, or have been incompletely characterized with respect to the role of developmental signaling pathways (*Lippmann et al., 2020*; *Workman and Svendsen, 2020*; *Lu et al., 2021*). As a potential alternative, hPSCs can also be used to generate immature, naïve endothelial progenitors (*Lian et al., 2014*) that could be used to better explore the induction of BBB phenotypes. For example, we recently reported that extended culture of such hPSC-derived endothelial progenitors in a minimal medium yielded ECs with improved BBB tight junction protein expression and localization, which led to improved paracellular barrier properties (*Nishihara et al., 2020*). However, as shown below, these cells exhibit high expression of PLVAP and little expression of GLUT-1, indicating the need for additional cues to drive CNS EC specification.

In this work, we aimed to define the effects of activating Wnt/β-catenin signaling in hPSC-derived, naïve endothelial progenitors and assess the extent to which this strategy would drive development of a CNS EC-like phenotype. We found that many aspects of the CNS EC phenotype, including the canonical GLUT-1, claudin-5, and PLVAP expression effects, were regulated by CHIR 99021, a small molecule agonist of Wnt/β-catenin signaling. CHIR treatment in matured ECs produced a more limited response. Whole-transcriptome analysis revealed definitive endothelial identity of the resulting cells and CHIR-upregulated expression of known CNS EC transcripts, including *LEF1*, *APCDD1*, *AXIN2*, *SLC2A1*, *CLDN5*, *LSR*, *ABCG2*, *SOX7*, and *ZIC3*. We also observed an unexpected CHIR-mediated upregulation of caveolin-1, which did not, however, correlate with increased uptake of a dextran tracer. Thus, we provide evidence that Wnt activation in hPSC-derived naïve endothelial progenitors is sufficient to induce many aspects of the CNS barrier EC phenotype, and we establish a model system for further systematic investigation of putative barriergenic cues.

## Results

### Wnt activation in hPSC-derived endothelial progenitors

We adapted an existing protocol to produce endothelial progenitor cells (EPCs) from hPSCs (*Lian et al., 2014*; *Bao et al., 2016*; *Figure 1A*). To achieve mesoderm specification, this method employs an initial activation of Wnt/β-catenin signaling with CHIR 99021 (CHIR), a small molecule inhibitor of glycogen synthase kinase-3 (GSK-3), which results in inhibition of GSK-3β-mediated β-catenin degradation. After 5 days of expansion, the resulting cultures contained a mixed population of CD34$^+$CD31$^+$ EPCs and CD34$^-$CD31$^-$ non-EPCs (*Figure 1B and C*). We used magnetic-activated cell sorting (MACS) to isolate CD31$^+$ cells from this mixed culture and plated these cells on collagen IV-coated plates in a minimal EC medium termed hECSR (*Nishihara et al., 2020*). We first asked whether Wnt3a, a ligand widely used to activate canonical Wnt/β-catenin signaling (*Kim et al., 2005*; *Kim et al., 2008*; *Liebner et al., 2008*; *Cecchelli et al., 2014*; *Praça et al., 2019*), could induce GLUT-1 expression in the resulting ECs. After 6 days of treatment, we observed a significant increase in the fraction of GLUT-1$^+$ ECs in Wnt3a-treated cultures compared to controls (*Figure 1D and E*). Consistent with previous observations (*Nishihara et al., 2020*), we also detected a population of calponin$^+$ smooth muscle protein 22-α$^+$ putative smooth muscle-like cells (SMLCs) outside the endothelial colonies

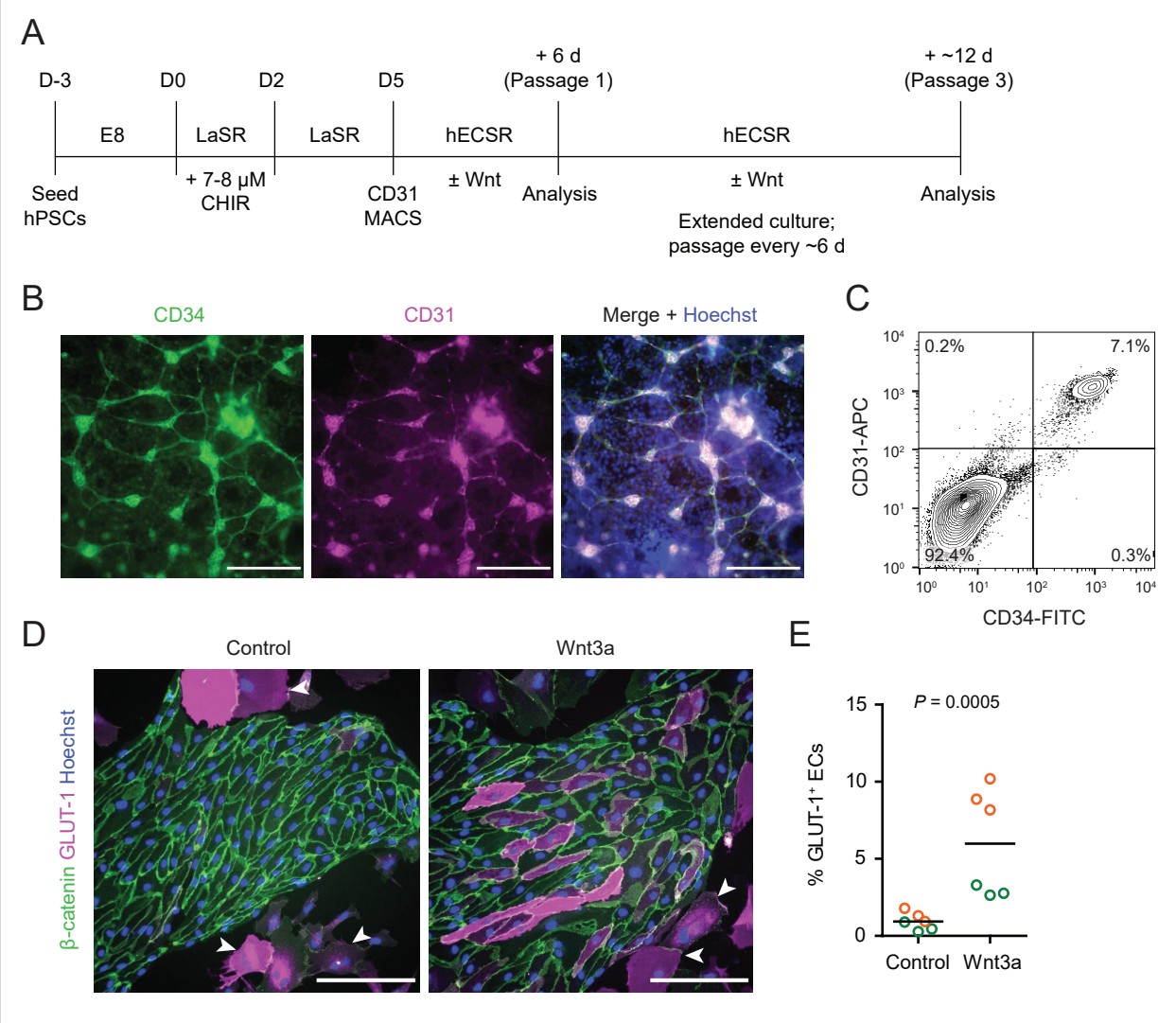

**Figure 1.** Human pluripotent stem cell (hPSC)-derived endothelial progenitors as a model for studying Wnt-mediated barriergenesis. (**A**) Overview of the endothelial differentiation and Wnt treatment protocol. (**B**) Immunocytochemistry analysis of CD34 and CD31 expression in D5 endothelial progenitor cells (EPCs) prior to magnetic-activated cell sorting (MACS). Hoechst nuclear counterstain is overlaid in the merged image. Scale bars: 200 μm. (**C**) Flow cytometry analysis of CD34 and CD31 expression in D5 EPCs prior to MACS. (**D**) Immunocytochemistry analysis of β-catenin and GLUT-1 expression in Passage 1 ECs treated with Wnt3a or control. Hoechst nuclear counterstain is overlaid. Arrowheads indicate smooth muscle-like cells (SMLCs). Scale bars: 200 μm. (**E**) Quantification of the percentage of GLUT-1$^+$ ECs in control- and Wnt3a-treated conditions. Points represent replicate wells from two independent differentiations of the IMR90-4 line, each differentiation indicated with a different color. Bars indicate mean values. p-value: two-way ANOVA.

The online version of this article includes the following figure supplement(s) for figure 1:

**Figure supplement 1.** Smooth muscle-like cells (SMLCs).

(*Figure 1—figure supplement 1*), and these SMLCs expressed GLUT-1 in both control and Wnt3a-treated conditions (*Figure 1D*).

Based on these promising results with Wnt3a, we next tested a low concentration (4 μM) of the GSK-3 inhibitor CHIR because of its ability to activate Wnt signaling in a receptor/co-receptor-independent manner. In addition to GLUT-1, we evaluated expression of two other key proteins: claudin-5, which is known to be upregulated in CNS ECs in response to Wnt (*Benz et al., 2019*), and caveolin-1, given the low rate of caveolin-mediated transcytosis in CNS compared to non-CNS ECs (*Reese and Karnovsky, 1967*; *Andreone et al., 2017*; *Figure 2A*). 4 μM CHIR robustly induced GLUT-1 expression in approximately 90% of ECs while increasing EC number and increasing EC purity to nearly 100% (*Figure 2B*). Furthermore, CHIR led to an approximately 1.5-fold increase in average

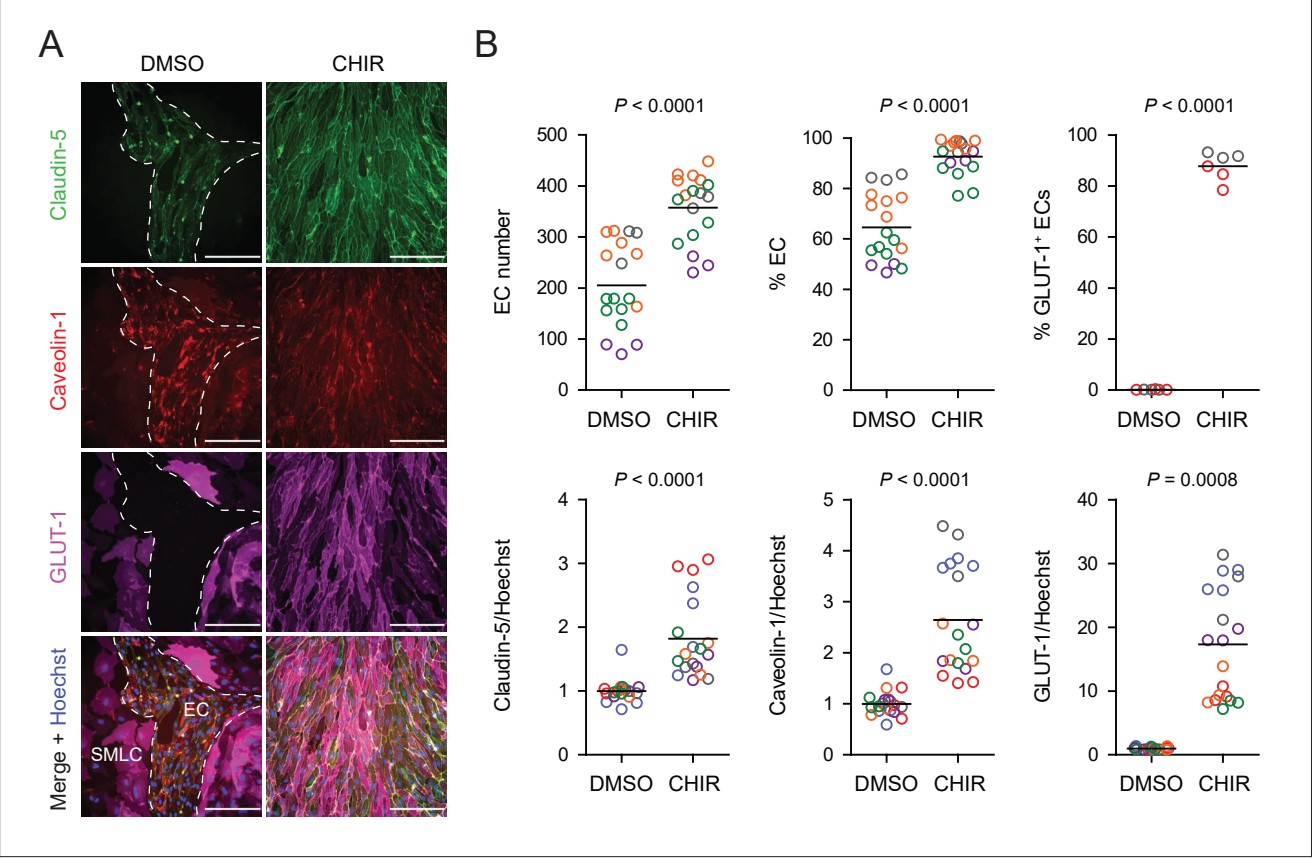

**Figure 2.** Effect of CHIR on endothelial properties. (**A**) Immunocytochemistry analysis of claudin-5, caveolin-1, and GLUT-1 expression in Passage 1 endothelial cells (ECs) treated with DMSO or 4 µM CHIR. Hoechst nuclear counterstain is overlaid in the merged images. Dashed lines indicate the border between an EC colony and smooth muscle-like cells (SMLCs) in the DMSO condition. Scale bars: 200 µm. (**B**) Quantification of images from the conditions described in (**A**) for number of ECs per 20× field, percentage of ECs (claudin-5+ cells relative to total nuclei), percentage of GLUT-1+ ECs (relative to total claudin-5+ ECs), and mean fluorescence intensity of claudin-5, caveolin-1, and GLUT-1 normalized to Hoechst mean fluorescence intensity within the area of claudin-5+ ECs only. Points represent replicate wells from 2 to 6 independent differentiations of the IMR90-4 line, each differentiation indicated with a different color. Bars indicate mean values. For the fluorescence intensity plots, values were normalized within each differentiation such that the mean of the DMSO condition equals 1. p-values: two-way ANOVA on unnormalized data.

The online version of this article includes the following figure supplement(s) for figure 2:

**Figure supplement 1.** Dose-dependent effects of CHIR on endothelial properties.

**Figure supplement 2.** CHIR-mediated effects in an additional human pluripotent stem cell (hPSC) line.

**Figure supplement 3.** β-catenin-dependence of CHIR-mediated GLUT-1 induction.

**Figure supplement 4.** Effect of CHIR on endothelial cell proliferation.

claudin-5 abundance and a 10- to 30-fold increase in GLUT-1 abundance, but also a 2- to 4-fold increase in caveolin-1 (***Figure 2B***). We therefore titrated CHIR to determine an optimal concentration for EC expansion, purity, GLUT-1 induction, and claudin-5 upregulation while limiting the undesirable non-CNS-like increase in caveolin-1 abundance. Although 2 µM CHIR did not lead to increased caveolin-1 expression compared to vehicle control (DMSO), it also did not elevate claudin-5 or GLUT-1 expression compared to control and was less effective in increasing EC number and EC purity than 4 µM CHIR (***Figure 2—figure supplement 1***). On the other hand, 6 µM CHIR further increased GLUT-1 abundance but also further increased caveolin-1 abundance and did not improve EC number, EC purity, or claudin-5 expression (***Figure 2—figure supplement 1***). Therefore, we conducted further experiments using 4 µM CHIR. We confirmed that the CHIR-mediated increases in EC purity, EC number, and caveolin-1 and GLUT-1 expression were conserved in an additional hPSC line, although claudin-5 upregulation was not apparent (***Figure 2—figure supplement 2***). We also used two hPSC lines with doxycycline-inducible expression of short hairpin RNAs targeting *CTNNB1* (β-catenin) to confirm that

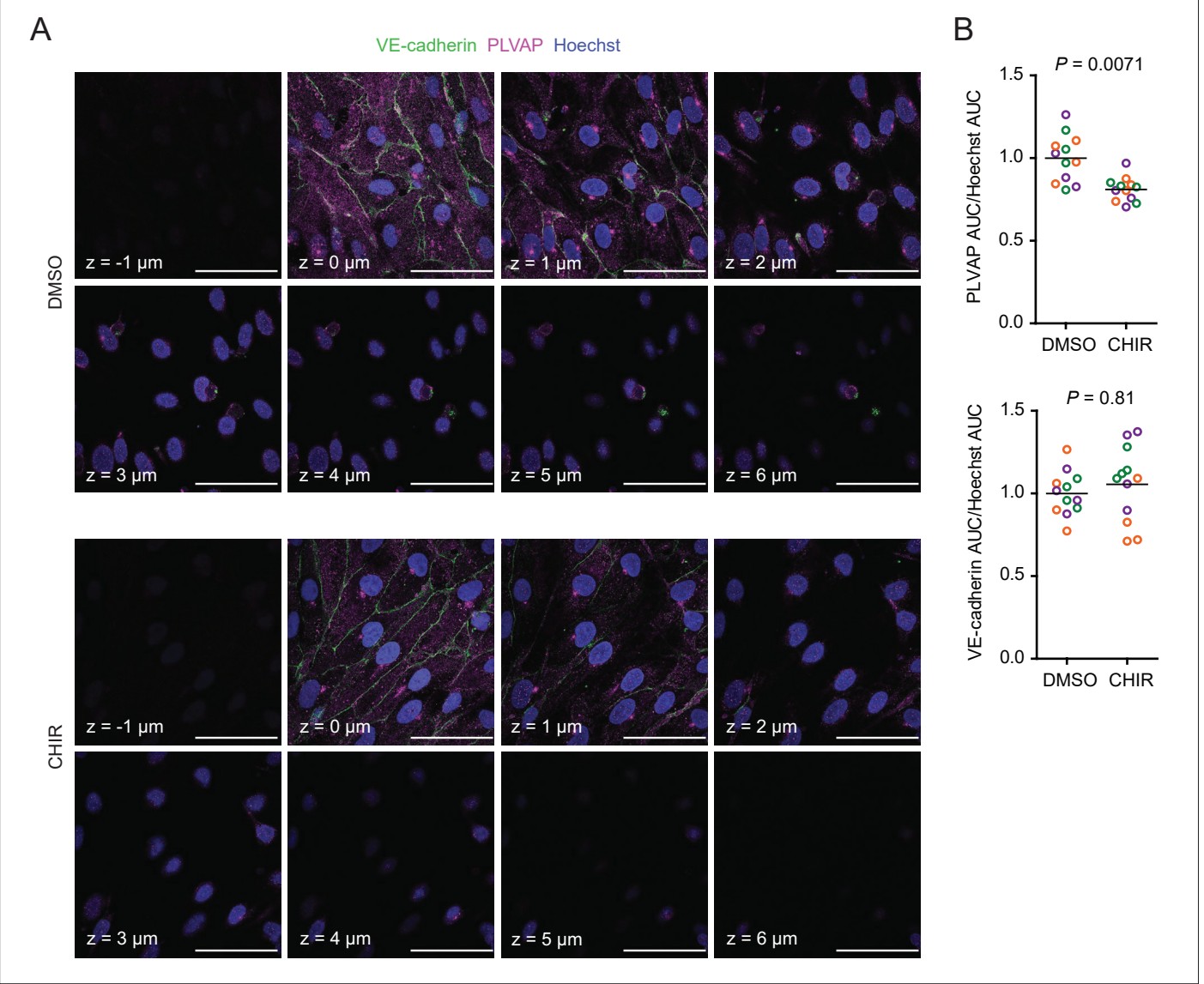

**Figure 3.** Effect of CHIR on endothelial PLVAP expression. (**A**) Confocal immunocytochemistry analysis of VE-cadherin and PLVAP expression in Passage 1 endothelial cells (ECs) treated with DMSO or CHIR. Hoechst nuclear counterstain is overlaid. Eight serial confocal Z-slices with 1 µm spacing are shown. Scale bars: 50 µm. (**B**) Quantification of PLVAP and VE-cadherin area under the curve (AUC) of mean fluorescence intensity versus Z-position normalized to Hoechst AUC. Points represent replicate wells from three independent differentiations of the IMR90-4 line, each differentiation indicated with a different color. Bars indicate mean values, with values normalized within each differentiation such that the mean of the DMSO condition equals 1. p-values: two-way ANOVA on unnormalized data.

CHIR-mediated upregulation of GLUT-1 in ECs was β-catenin-dependent. Indeed, doxycycline treatment in combination with CHIR significantly reduced GLUT-1 abundance in ECs derived from these hPSC lines (*Figure 2—figure supplement 3*). Finally, we confirmed that increased EC number was the result of increased EC proliferation in CHIR-treated cultures (*Figure 2—figure supplement 4*). Together, these results suggest that activation of the Wnt/β-catenin pathway is capable of inducing CNS-like phenotypes in hPSC-derived endothelial progenitors.

## Effects of CHIR-mediated Wnt activation in endothelial progenitors

Since CHIR elicited a robust Wnt-mediated response, we next asked whether other aspects of the CNS EC barrier phenotype were CHIR-regulated. PLVAP, a protein that forms bridges across both caveolae and fenestrae (*Herrnberger et al., 2012*), is one such canonically Wnt-downregulated

protein. We therefore first evaluated PLVAP expression in Passage 1 control (DMSO) or CHIR-treated ECs using confocal microscopy (*Figure 3A*). We observed numerous PLVAP+ punctate vesicle-like structures in both conditions, with CHIR treatment reducing PLVAP abundance by approximately 20% (*Figure 3A and B*). This effect was not apparent in western blots of Passage 1 ECs, likely due to the relatively modest effect (*Figure 4A and B*). However, after two more passages (*Figure 1A*), Passage 3 ECs demonstrated a robust downregulation of PLVAP in CHIR-treated cells compared to controls (*Figure 4C and D*). We also used western blotting to confirm CHIR-mediated upregulation of GLUT-1 and claudin-5 both at Passage 1 and Passage 3 (*Figure 4A–D*). We next evaluated expression of the tricellular tight junction protein LSR (angulin-1) because of its enrichment in CNS versus non-CNS ECs, and the temporal similarity between LSR induction and the early stage of Wnt-mediated CNS barri-ergenesis (*Sohet et al., 2015*). We found that CHIR treatment led to a strong increase in LSR expression in both Passage 1 and Passage 3 ECs (*Figure 4A–D*), suggesting that Wnt signaling upregulates multiple necessary components of the CNS EC bicellular and tricellular junctions.

CHIR treatment produced two apparently competing changes in ECs related to vesicular transport: an expected downregulation of PLVAP and an unexpected upregulation of caveolin-1. We therefore asked whether the rate of total fluid-phase endocytosis differed between CHIR-treated and control ECs using a fluorescently labeled 10 kDa dextran as a tracer. After incubating Passage 1 cultures with dextran for 2 hr at 37°C, we used flow cytometry to gate CD31+ ECs and assess total dextran accu-mulation (*Figure 5A and B*). In ECs incubated at 37°C, CHIR treatment did not change the geometric mean dextran signal compared to DMSO (*Figure 5B and C*), but did cause a broadening of the distri-bution of dextran intensities as quantified by the coefficient of variation (CV), indicative of subpopula-tions of cells with decreased and increased dextran uptake (*Figure 5B and D*). We confirmed that the dextran signal measured by this assay was endocytosis-dependent by carrying out the assay at 4°C and with inhibitors of specific endocytic pathways (*Figure 5—figure supplement 1A–C*). Compared to vehicle control, chlorpromazine (inhibitor of clathrin-mediated endocytosis) and rottlerin (inhibitor of macropinocytosis) both decreased dextran uptake, while nystatin (inhibitor of caveolin-mediated endocytosis) did not significantly affect uptake (*Figure 5—figure supplement 1B and C*), consistent with the very small number of dextran+ caveolin-1+ puncta observed by confocal imaging (*Figure 5—figure supplement 1D*). Thus, despite the generally uniform elevation of caveolin-1 and decrease of PLVAP observed by immunocytochemistry in CHIR-treated ECs, our functional assay suggests neither an overall increase nor decrease in total fluid-phase endocytosis. Instead, it indicates that CHIR increases the heterogeneity of the EC population with respect to the rate of endocytosis.

We also compared the paracellular barrier properties of DMSO- and CHIR-treated ECs. Because Passage 1 cultures contain SMLCs that preclude formation of a confluent endothelial monolayer, we evaluated paracellular barrier properties of Passage 3 ECs that had undergone selective dissociation and replating (see Materials and methods), a strategy that effectively purifies the cultures (*Nishihara et al., 2020*). CHIR-treated Passage 3 ECs had elevated transendothelial electrical resistance (TEER) (*Figure 5E*) and decreased permeability to the small molecule tracer sodium fluorescein (*Figure 5F*). Together, these results are consistent with CHIR-mediated increases to tight junction protein expres-sion (e.g., claudin-5 and LSR) and suggest that Wnt activation leads to functional improvements to paracellular barrier in this system.

Given the relatively weak responses to Wnt activation in adult mouse liver ECs in vivo (*Munji et al., 2019*) and adult mouse brain ECs cultured in vitro (*Sabbagh and Nathans, 2020*), we sought to deter-mine whether the immature, potentially more plastic state of hPSC-derived endothelial progenitors contributed to the relatively robust CHIR-mediated response we observed. To test this hypothesis, we matured hPSC-derived ECs in vitro for four passages (until approximately day 30) prior to initi-ating CHIR treatment for 6 days and compared the resulting cells to differentiation-matched samples treated with CHIR immediately after MACS (*Figure 6A*). Both Passage 1 DMSO-treated ECs and Passage 5 DMSO-treated ECs, which are analogous to EECM-BMEC-like cells we previously reported (*Nishihara et al., 2020*), did not have detectable GLUT-1 expression (*Figure 6B*). Compared to DMSO controls, the CHIR-treated Passage 5 ECs exhibited no increase in GLUT-1 abundance (*Figure 6B–D*), which contrasts with the marked increase observed when CHIR treatment was initiated immediately after MACS (*Figure 6B–D*). Furthermore, CHIR treatment in matured ECs did not increase claudin-5 expression and did not increase EC number (*Figure 6B–D*), in contrast to the increases observed in both properties when treatment was initiated immediately after MACS (*Figure 6B–D*). We observed

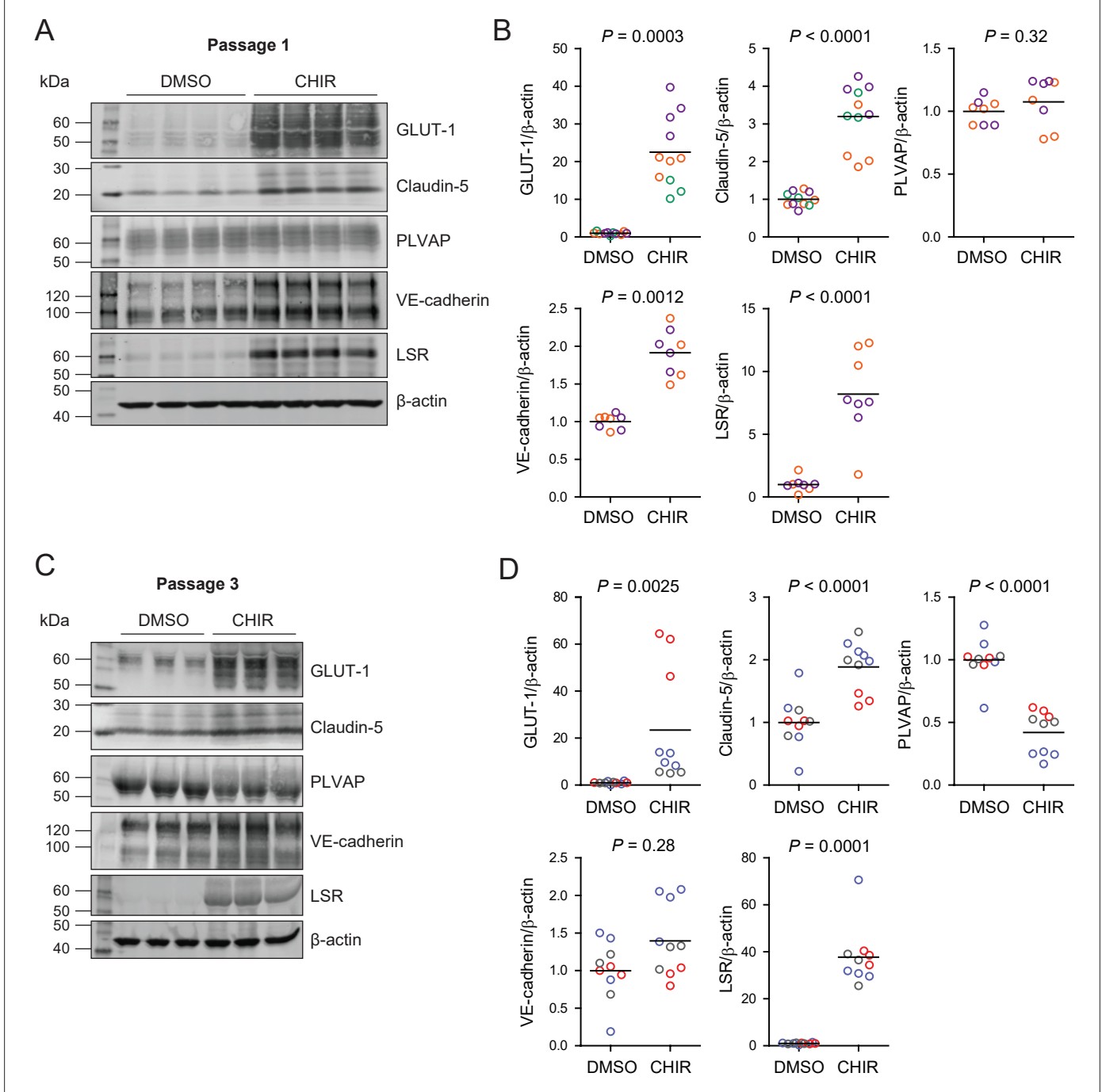

**Figure 4.** Effect of CHIR on protein expression in Passage 1 and Passage 3 endothelial cells (ECs). (**A**) Western blots of Passage 1 ECs treated with DMSO or CHIR probed for GLUT-1, claudin-5, PLVAP, VE-cadherin, LSR, and β-actin. (**B**) Quantification of western blots of Passage 1 ECs. GLUT-1, claudin-5, PLVAP, VE-cadherin, and LSR band intensities was normalized to β-actin band intensity. Points represent replicate wells from 2 to 3 independent differentiations of the IMR90-4 line, each differentiation indicated with a different color. Bars indicate mean values, with values normalized within each differentiation such that the mean of the DMSO condition equals 1. p-values: two-way ANOVA on unnormalized data. (**C**) Western blots of Passage 3 ECs treated with DMSO or CHIR probed for GLUT-1, claudin-5, PLVAP, VE-cadherin, LSR, and β-actin. (**D**) Quantification of western blots of Passage 3 ECs. GLUT-1, claudin-5, PLVAP, VE-cadherin, and LSR band intensities was normalized to β-actin band intensity. Points represent replicate wells from three independent differentiations of the IMR90-4 line, each differentiation indicated with a different color. Bars indicate mean values, with values normalized within each differentiation such that the mean of the DMSO condition equals 1. p-values: two-way ANOVA on unnormalized data.

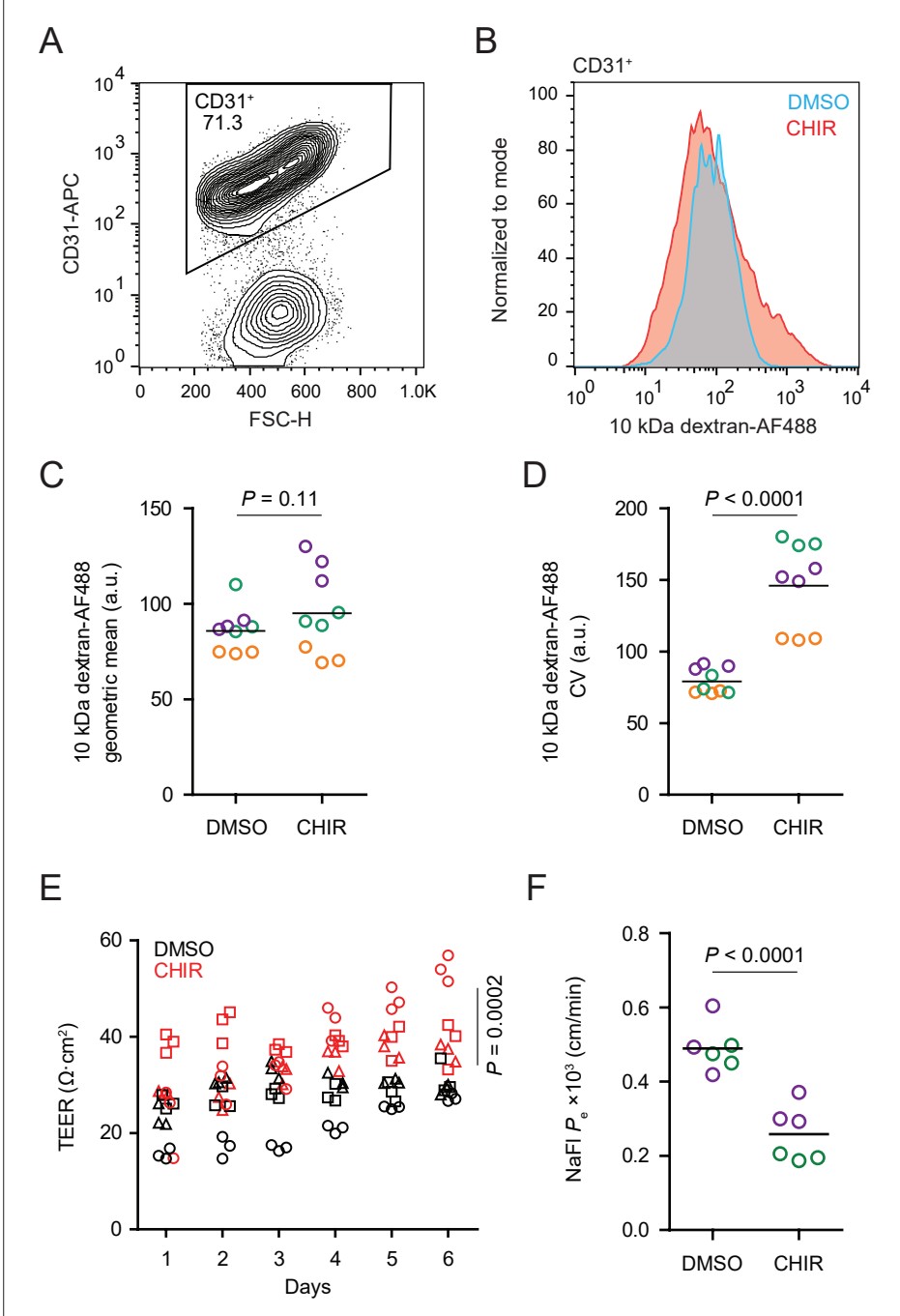

**Figure 5.** Functional properties of CHIR- and DMSO-treated endothelial cells (ECs). (**A**) Flow cytometry analysis of CD31 expression in Passage 1 ECs following the dextran internalization assay. CD31+ cells were gated for further analysis. (**B**) Flow cytometry analysis of 10 kDa dextran-Alexa Fluor 488 (AF488) abundance in CD31+ cells. Cells were treated with DMSO or CHIR for 6 days prior to the assay. Representative plots from cells incubated with dextran for 2 hr at 37°C are shown. (**C**) Quantification of 10 kDa dextran-AF488 geometric mean fluorescence intensity in CD31+ cells. Treatment and assay conditions were as described in (**B**). Points represent replicate wells from three independent differentiations of the IMR90-4 line, each differentiation indicated with a different color. Bars indicate mean values. p-value: two-way ANOVA. (**D**) Quantification of the coefficient of variation (CV) of 10 kDa dextran-AF488 fluorescence intensity in CD31+ cells. Points represent replicate wells from three independent differentiations of the IMR90-4 line, each differentiation indicated with a different color. Bars indicate mean values. p-value: two-way ANOVA. (**E**) Transendothelial electrical resistance (TEER) of Passage 3 ECs. The x-axis indicates the number of days after seeding cells on Transwell inserts. Points represent replicate wells from

*Figure 5 continued on next page*

*Figure 5 continued*

three independent differentiations of the IMR90-4 line, each differentiation indicated with a different shape. p-value: two-way ANOVA. (**F**) Permeability of Passage 3 ECs to sodium fluorescein. Points represent replicate wells from two independent differentiations of the IMR90-4 line, each differentiation indicated with a different color. Bars indicate mean values. p-value: two-way ANOVA.

The online version of this article includes the following figure supplement(s) for figure 5:

**Figure supplement 1.** Endocytosis dependence of dextran uptake.

a similar lack of robust GLUT-1 induction in an additional differentiation and an additional hPSC line in which CHIR treatment was carried out at Passage 4 (*Figure 6—figure supplement 1*). Together, these data suggest that early, naïve endothelial progenitors are more responsive to Wnt activation than more mature ECs derived by the same differentiation protocol.

## Comprehensive profiling of the Wnt-regulated endothelial transcriptome

We turned next to RNA-sequencing as an unbiased method to assess the impacts of Wnt activation on the EC transcriptome. We performed four independent differentiations and analyzed Passage 1 ECs treated with DMSO or CHIR using fluorescence-activated cell sorting (FACS) to isolate CD31$^+$ ECs from the mixed EC/SMLC cultures. We also sequenced the SMLCs from DMSO-treated cultures at Passage 1 from two of these differentiations. DMSO- and CHIR-treated ECs at Passage 3 from three of these differentiations were also sequenced. Principal component analysis of the resulting whole-transcriptome profiles revealed that the two cell types (ECs and SMLCs) segregated along principal component (PC) 1, which explained 57% of the variance. In ECs, the effects of passage number and treatment were reflected in PC 2, which explained 21% of the variance (*Figure 7A*). We next validated the endothelial identity of our cells; we observed that canonical endothelial marker genes (including *CDH5, CD34, PECAM1, CLDN5, ERG,* and *FLI1*) were enriched in ECs compared to SMLCs and had high absolute abundance, on the order of 100–1000 transcripts per million (TPM) (*Figure 7B*, *Supplementary file 1*). SMLCs expressed mesenchymal (mural/fibroblast)-related transcripts (including *PDGFRB, CSPG4, PDGFRA, TBX2, CNN1,* and *COL1A1*), which ECs generally lacked, although we did observe slight enrichment of some of these genes in Passage 1 DMSO-treated ECs, likely reflective of a small amount of SMLC contamination despite CD31 FACS (*Figure 7B*). SMLCs also expressed *SLC2A1* (*Supplementary file 1*) consistent with protein-level observations (*Figure 1D*). We also observed little to no expression of the epithelial genes *CDH1, EPCAM, CLDN1, CLDN3* (*Castro Dias et al., 2019*), *CLDN4,* and *CLDN6*, reflecting the definitive endothelial nature of the cells (*Figure 7B*, *Supplementary file 1*).

First comparing CHIR- and DMSO-treated ECs at Passage 1, we identified 1369 significantly upregulated genes and 2037 significantly downregulated genes (*Figure 7C*, *Supplementary file 2*). CHIR-upregulated genes included *SLC2A1, CLDN5, LSR,* and *CAV1*, consistent with protein-level assays. *PLVAP* was downregulated, as were a number of mesenchymal genes (*TAGLN, COL1A1*), again reflective of slight contamination of SMLC transcripts in the DMSO-treated EC samples (*Figure 7C and D*). Additionally, important downstream effectors of Wnt signaling were upregulated, including the transcription factors *LEF1* and *TCF7*, the negative regulator *AXIN2*, and the negative regulator *APCDD1*, which is known to modulate Wnt-regulated barriergenesis in retinal endothelium (*Mazzoni et al., 2017*; *Figure 7C and D*). We also identified upregulated transcription factors: *ZIC3,* which is highly enriched in brain and retinal ECs in vivo and downstream of Frizzled4 signaling (*Wang et al., 2012*; *Sabbagh et al., 2018*), and *SOX7,* which acts cooperatively with *SOX17* and *SOX18* in retinal angiogenesis (*Zhou et al., 2015*), were upregulated by CHIR in our system (*Figure 7D*). *MSX1* and *EBF1,* which are expressed by murine brain ECs in vivo (*Vanlandewijck et al., 2018*), were also CHIR-upregulated (*Figure 7D*). Additional CHIR-upregulated genes included *ABCG2* (encoding the efflux transporter breast cancer resistance protein [BCRP]), *APLN,* a tip cell marker enriched in postnatal day 7 murine brain ECs compared to those of other organs, and subsequently downregulated in adulthood (*Sabbagh et al., 2018*; *Sabbagh and Nathans, 2020*), and *FLVCR2,* a disease-associated gene with a recently identified role in brain angiogenesis (*Santander et al., 2020*; *Figure 7C and D*). Finally, we detected CHIR-mediated downregulation of the fatty acid-binding protein-encoding

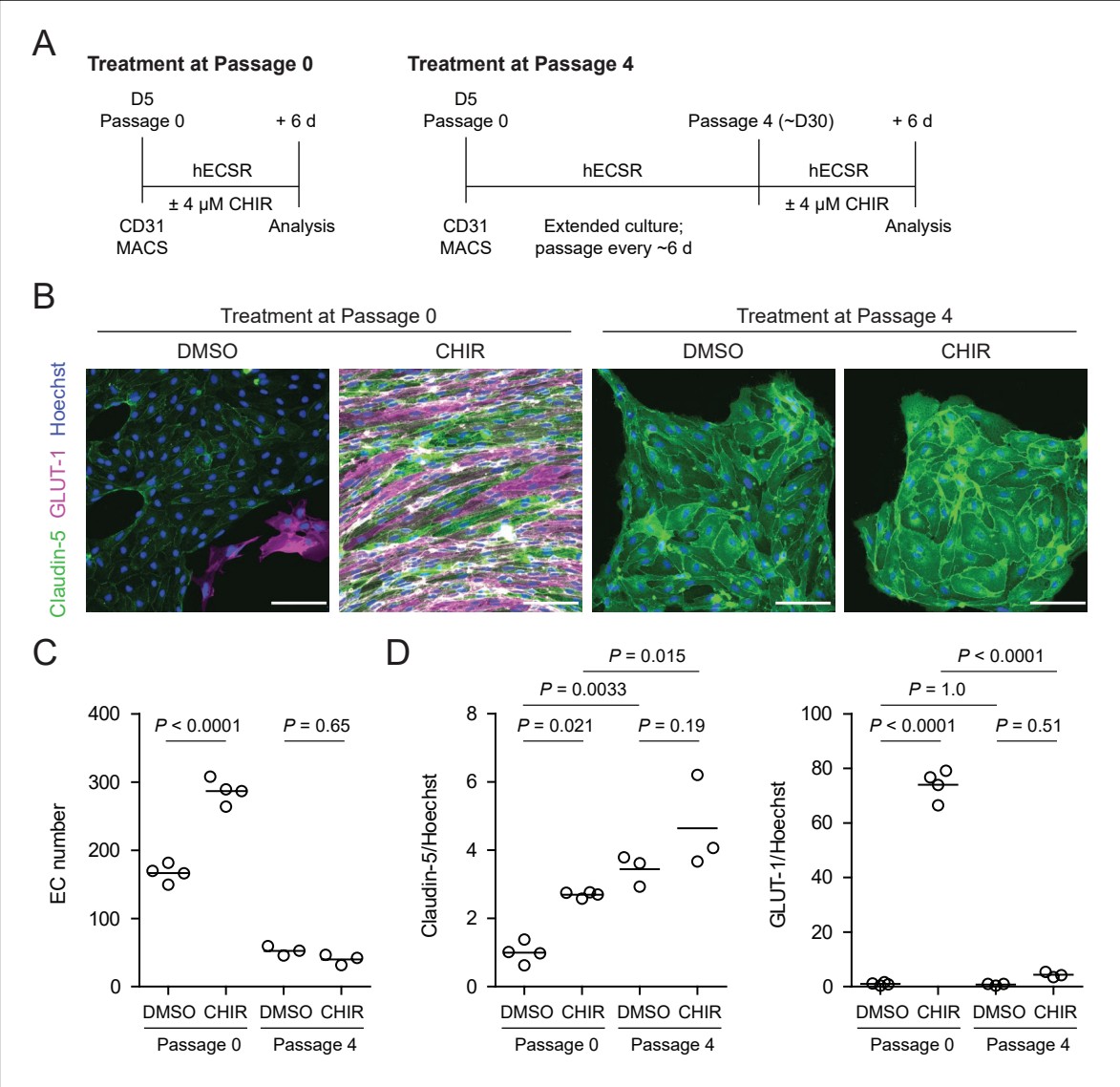

**Figure 6.** Effect of CHIR treatment in endothelial progenitor cells (EPCs) and matured endothelium. (**A**) Overview of the endothelial differentiation, extended culture, and CHIR treatment protocols. (**B**) Immunocytochemistry analysis of claudin-5 and GLUT-1 expression in endothelial cells (ECs) treated with DMSO or CHIR as outlined in (**A**). Images from the IMR90-4 line are shown. Hoechst nuclear counterstain is overlaid. Scale bars: 100 μm. (**C**) Quantification of images from the conditions described in (**B**) for number of ECs per 30× field. Points represent replicate wells from one differentiation of the IMR90-4 line. Bars indicate mean values. p-values: ANOVA followed by Tukey's honest significant difference (HSD) test. (**D**) Quantification of images from the conditions described in (**B**) for GLUT-1 and claudin-5 mean fluorescence intensity normalized to Hoechst mean fluorescence intensity within the area of claudin-5+ ECs only. Points represent replicate wells from one differentiation of the IMR90-4 line. Bars indicate mean values, with values normalized such that the mean of the DMSO condition equals 1. p-values: ANOVA followed by Tukey's HSD test.

The online version of this article includes the following figure supplement(s) for figure 6:

**Figure supplement 1.** Effect of CHIR treatment in matured endothelium.

*FABP4*, which is depleted in brain ECs compared to those of peripheral organs (*Sabbagh et al., 2018*). We also observed similar downregulation of *SMAD6*, which is depleted in brain ECs compared to lung ECs and is a putative negative regulator of BMP-mediated angiogenesis (*Mouillesseaux et al., 2016*; *Vanlandewijck et al., 2018*; *Figure 7D*).

In Passage 3 ECs, many of the CHIR-mediated gene expression changes observed at Passage 1 persisted, including *SLC2A1*, *LSR*, *LEF1*, *AXIN2*, *APCDD1*, *ZIC3*, *EBF1*, *FLVCR2*, and *ABCG2* upregulation and *PLVAP* downregulation (*Figure 7E*, *Figure 7—figure supplement 1*). Additional concordantly CHIR-upregulated genes encoding secreted factors, transcription factors, and transmembrane

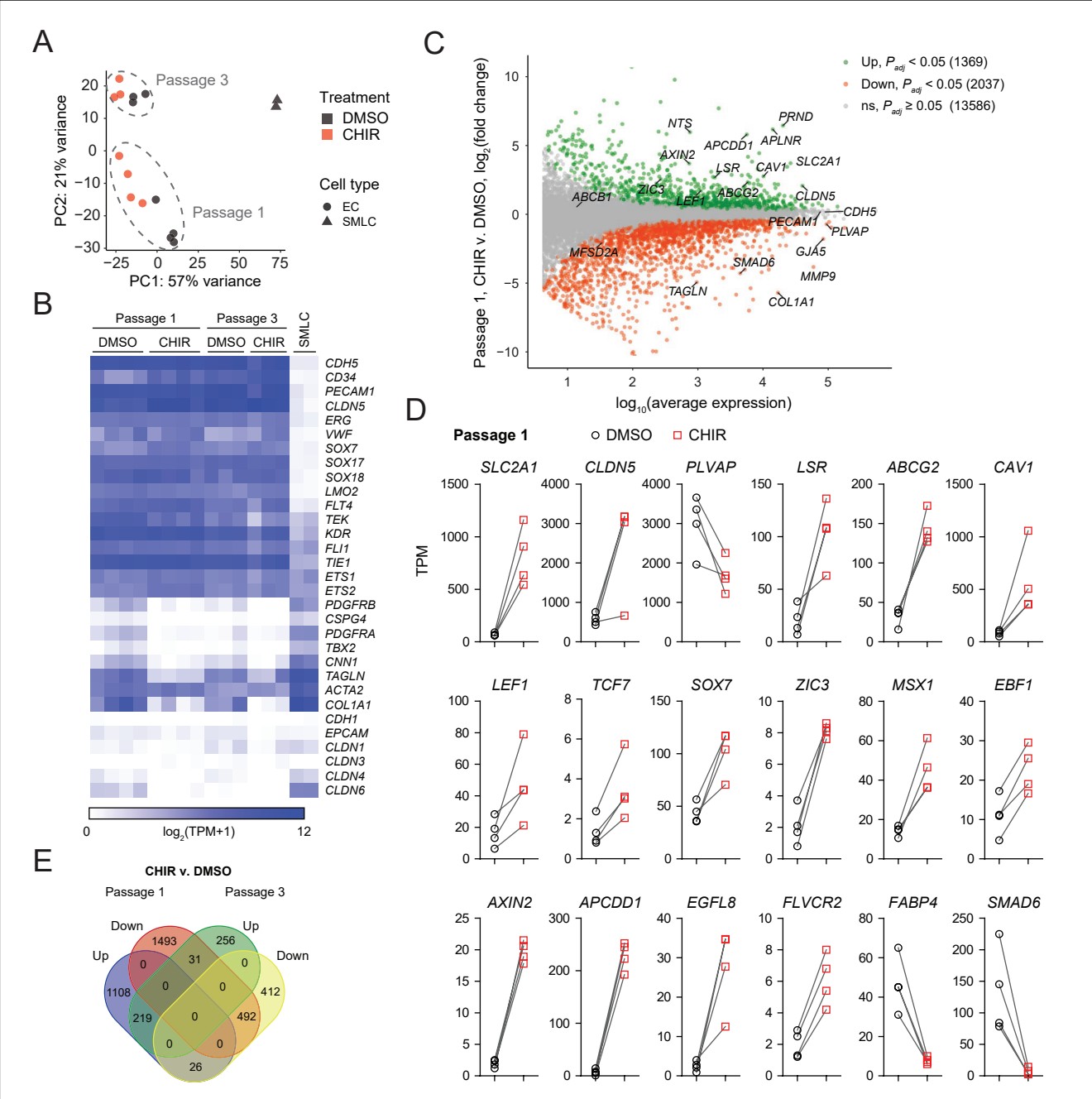

**Figure 7.** RNA-seq of DMSO- and CHIR-treated endothelial cells (ECs). (**A**) Principal component (PC) analysis of EC and smooth muscle-like cell (SMLC) whole-transcriptome data subject to variance stabilizing transformation by DESeq2. Points from Passage 1 ECs represent cells from four independent differentiations of the IMR90-4 line, points from Passage 3 ECs represent cells from three independent differentiations of the IMR90-4 line, and points from SMLCs represent two independent differentiations of the IMR90-4 line. Points are colored based on treatment: DMSO (black), CHIR (red). Data are plotted in the space of the first two PCs, with the percentage of variance explained by PC1 and PC2 shown in axis labels. Dashed lines indicate points from Passage 1 and Passage 3 ECs, and are not confidence ellipses. (**B**) Heatmap of transcript abundance [$\log_2$(TPM + 1)] for endothelial, mesenchymal, and epithelial genes across all samples. Abundance data for all transcripts are provided in ***Supplementary file 1***. (**C**) Differential expression analysis of Passage 1 CHIR-treated ECs compared to Passage 1 DMSO-treated ECs. Differentially expressed genes (adjusted p-values<0.05, DESeq2 Wald test with Benjamini–Hochberg correction) are highlighted in green (upregulated) and red (downregulated). The number of upregulated, downregulated, and nonsignificant (ns) genes is shown in the legend. Complete results of differential expression analysis are provided in ***Supplementary file 2***. (**D**) Transcript abundance (TPM) of Wnt-regulated, barrier-related genes in Passage 1 DMSO- and CHIR-treated ECs. Points represent cells from four independent differentiations of the IMR90-4 line and lines connect points from matched differentiations. All genes shown were differentially expressed (adjusted p-values<0.05, DESeq2 Wald test with Benjamini–Hochberg correction). p-values are provided in ***Supplementary file 2***. (**E**) Venn diagram

*Figure 7 continued on next page*

*Figure 7 continued*

illustrating the number of genes identified as upregulated or downregulated (adjusted p-values<0.05, DESeq2 Wald test with Benjamini–Hochberg correction) in ECs treated with CHIR versus DMSO at Passage 1 compared to Passage 3. Gene lists are provided in *Supplementary file 2*, and selected genes are shown in *Figure 7—figure supplement 1* and *Figure 7—figure supplement 2*.

The online version of this article includes the following figure supplement(s) for figure 7:

**Figure supplement 1.** Differential expression analysis of Passage 3 endothelial cells (ECs) treated with CHIR versus DMSO.

**Figure supplement 2.** Genes upregulated by CHIR at both Passage 1 and Passage 3.

**Figure supplement 3.** Gene correlation network analysis.

**Figure supplement 4.** Differential expression analysis of Passage 1 versus Passage 3 endothelial cells (ECs).

**Figure supplement 5.** Blood–brain barrier transcriptional profile.

**Figure supplement 6.** Expression of Wnt pathway components in naïve endothelial cells (ECs).

proteins are shown in *Figure 7—figure supplement 2* and include *REEP1*, a gene enriched in brain versus non-brain ECs (*Sabbagh et al., 2018*; *Vanlandewijck et al., 2018*) that encodes a regulator of endoplasmic reticulum function and the Notch ligand-encoding gene *JAG2*. On the other hand, at Passage 3, *CLDN5* was not upregulated in CHIR-treated cells compared to DMSO-treated cells, but was highly expressed (~2500 TPM). Similarly, *CAV1* abundance remained high, but was not CHIR-upregulated in Passage 3 cells (*Figure 7—figure supplement 1*). Conversely, *JAM2*, which encodes junctional adhesion molecule 2, a component of EC tight junctions (*Aurrand-Lions et al., 2001*; *Tietz and Engelhardt, 2015*), was upregulated by CHIR at Passage 3, but not at Passage 1, as was the retinol-binding protein-encoding gene *RBP1* (*Figure 7—figure supplement 1*).

We used weighted gene correlation network analysis (WGCNA) (*Zhang and Horvath, 2005*; *Langfelder and Horvath, 2008*) to identify modules containing genes with highly correlated expression across the 14 EC samples (*Figure 7—figure supplement 3A*, *Supplementary file 3*). One such module (the green module, containing 441 genes) had a representative gene expression profile (module eigengene) with a strong, positive correlation with CHIR treatment (*Figure 7—figure supplement 3B*). Importantly, genes central to this module included canonical transcriptional targets of Wnt/β-catenin signaling, including *AXIN2* and *APCDD1*, further supporting the key role of β-catenin signaling in transcriptional changes observed in CHIR-treated ECs. Additional central (highly correlated) genes within the green module included *SLC2A1*, *ZIC3*, and *FLVCR2*, consistent with pairwise differential expression analysis, transcription factors (*CASZ1*, *PRRX1*), and genes with putative roles in vesicle trafficking (*SNX4*, *ARL8B*, *AP1AR*, *VTI1A*, *VPS41*) and lipid metabolism (*AGPAT5*, *ASAH1*) (*Figure 7—figure supplement 3C*).

To determine the effects of extended culture, we next compared control (DMSO-treated) ECs at Passage 3 versus Passage 1 (*Figure 7—figure supplement 4*, *Supplementary file 2*). Extended culture to Passage 3 in the absence of exogeneous Wnt activation led to 1521 upregulated genes, including *CLDN5* and *CAV1*, consistent with previously reported protein-level observations in EECM-BMEC-like cells (*Nishihara et al., 2020*), which are analogous to Passage 3 DMSO-treated cells. We also observed 1625 downregulated genes, including marked downregulation of *PLVAP* (*Figure 7—figure supplement 4*). *SLC2A1*, however, was not upregulated at Passage 3 (*Figure 7—figure supplement 4*), concordant with absence of GLUT-1 protein expression in the control ECs (*Figure 6B*), nor was *LSR*. Further, despite some similarly regulated genes between the passage number and CHIR treatment comparisons (e.g., *CLDN5*, *CAV1*, *PLVAP*), the transcriptional responses to these two experimental variables were globally distinct as assessed by gene correlation network analysis (*Figure 7—figure supplement 3B*). We also evaluated transcript-level expression of components of the Wnt signaling pathway in Passage 3 control (DMSO-treated) ECs as a first step towards understanding the relative lack of responsiveness observed when CHIR treatment was initiated in matured (Passage 4) ECs (*Figure 7—figure supplement 4*). While *CTNNB1*, *GSK3B*, and genes encoding components of the destruction complex were not significantly different between Passage 3 and Passage 1, *LEF1* and *TCF7* were strongly downregulated in Passage 3 cells (*Figure 7—figure supplement 4*).

Finally, to further understand the strengths and limitations of this model system both as a readout of early developmental changes in CNS ECs (Passage 1 cells) or as a source of CNS-like ECs for use in downstream modeling applications, we evaluated absolute transcript abundance and effects of treatment or passage number on 53 characteristic CNS EC genes encompassing tight junction

components, vesicle trafficking machinery, solute carriers, and ATP-binding cassette (ABC) efflux transporters selected based on high expression in human brain ECs from a meta-analysis of single-cell RNA-seq data (*Gastfriend et al., 2021*; *Figure 7—figure supplement 5*). While ECs expressed *CLDN5*, *TJP1*, *TJP2*, *OLCN*, and *LSR*, they lacked *MARVELD2* (encoding tricellulin) under all conditions. ECs under all conditions also lacked *MFSD2A* and, despite CHIR-mediated downregulation of *PLVAP*, retained high absolute expression of this and other caveolae-associated genes. Finally, while many solute carriers and ABC transporters were expressed (*SLC2A1*, *SLC3A2*, *SLC16A1*, *SLC38A2*, *ABCG2*), others expressed at the in vivo human BBB were not (*SLC5A3*, *SLC7A11*, *SLC38A3*, *SLCO1A2*, *ABCB1*) (*Figure 7—figure supplement 5*). Thus, while CHIR treatment yields ECs with certain elements of CNS-like character, additional molecular signals are likely necessary to impart other aspects of the in vivo CNS EC phenotype.

## The Wnt-regulated endothelial transcriptome in multiple contexts

To globally assess whether CHIR-mediated gene expression changes in our system are characteristic of the responses observed in ECs in vivo and similar to those observed in other in vitro contexts, we compared our RNA-seq dataset to those of studies that employed a genetic strategy for β-catenin stabilization (the *Ctnnb1*flex3 allele) in adult mouse ECs in several contexts: (i) pituitary ECs, which acquire some BBB-like properties upon β-catenin stabilization *Wang et al., 2019*; (ii) liver ECs, which exhibit little to no barriergenic response to β-catenin stabilization (*Munji et al., 2019*); (iii) brain ECs briefly cultured in vitro, which rapidly lose their BBB-specific gene expression profile even with β-catenin stabilization (*Sabbagh and Nathans, 2020*), and offer the most direct comparison to our in vitro model system. Upon recombination, the *Ctnnb1*flex3 allele produces a dominant mutant β-catenin lacking residues that are phosphorylated by GSK-3β to target β-catenin for degradation (*Harada et al., 1999*); as such, this strategy for ligand- and receptor-independent Wnt activation by β-catenin stabilization is similar to CHIR treatment, although GSK-3 phosphorylates targets other than β-catenin (discussed below).

We first used literature RNA-seq data from postnatal day 7 murine brain, liver, lung, and kidney ECs (*Sabbagh et al., 2018*) to define core sets of genes in brain ECs that are differentially expressed compared to all three of the other organs (*Figure 8A and B*). Using the resulting sets of 1094 brain-enriched and 506 brain-depleted genes, we asked how many genes in our Passage 1 ECs were concordantly regulated by CHIR: 130 of the brain-enriched genes were CHIR-upregulated and 116 of the brain-depleted genes were CHIR-downregulated (*Figure 8C*). At Passage 3, 61 genes were concordantly upregulated and 46 downregulated (*Figure 8—figure supplement 1*). In pituitary ECs with β-catenin stabilization, 102 of the brain-enriched genes were upregulated and 48 of the brain-depleted genes were downregulated (*Figure 8D*). Compared with the pituitary ECs, there were far fewer concordantly regulated genes in liver ECs with β-catenin stabilization, with 25 upregulated and 1 downregulated (*Figure 8E*). Finally, cultured primary mouse brain ECs with β-catenin stabilization exhibited 72 concordantly upregulated and 16 downregulated genes (*Figure 8F*). The only gene concordantly regulated in all four comparisons was the canonical Wnt target *AXIN2*. Several additional genes were concordantly upregulated in three of four, including *TCF7*, *FAM107A*, *NKD1*, *TNFRSF19*, *GLUL*, *SLC30A1*, and *ABCB1*, which was the only gene concordantly regulated in all comparisons except the hPSC-derived ECs (*Figure 8G*). Several canonical target genes were shared by the hPSC-derived EC and pituitary EC systems, including *APCDD1*, *LEF1*, *CLDN5*, and *SLC2A1*; also in this category were *LSR*, the zinc/manganese transporter *SLC39A8*, and 12 additional genes (*Figure 8G*). Notably, the caveolae inhibitor *MFSD2A* was robustly upregulated by β-catenin in pituitary ECs, but not in any other context (*Figure 8C–F*), suggesting that other brain-derived factors may cooperate with Wnt to regulate expression of this important inhibitor of caveolin-mediated transcytosis. Complete gene lists from this comparative analysis are provided in *Supplementary file 4*. In sum, the data suggest that the hPSC-derived ECs responded to Wnt activation in a fashion that led to modest induction of CNS transcriptional programs and that the response was most similar to the pituitary β-catenin stabilization model. Importantly, this analysis also supports the hypothesis that immature endothelium is highly responsive to Wnt activation where mature (adult) endothelium is largely refractory except in regions proximal to barrier-forming regions.

Last, because GSK-3 is a component of numerous signaling pathways in addition to Wnt/β-catenin (*Eto et al., 2005*; *Beurel et al., 2015*; *Hermida et al., 2017*), we used RNA-seq data to infer

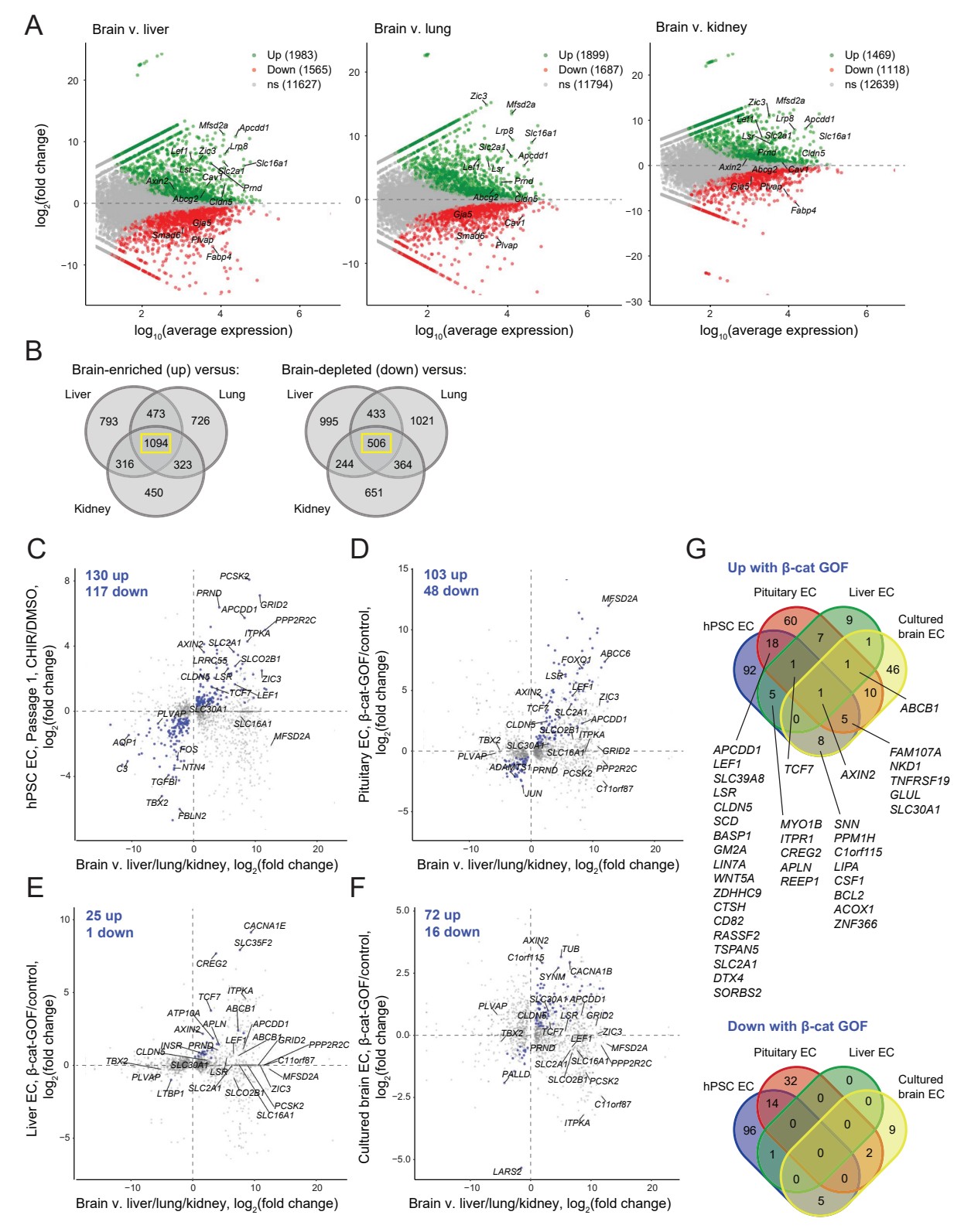

**Figure 8.** Identification of concordantly Wnt-regulated central nervous system (CNS) endothelial cell (EC)-associated genes in RNA-seq data. (**A**) Differential expression analysis of P7 murine brain ECs compared to liver, lung, or kidney ECs (*Sabbagh et al., 2018*). Differentially expressed genes (adjusted p-values<0.05, DESeq2 Wald test with Benjamini–Hochberg correction) are highlighted in green (up, brain-enriched) and red (down, brain-depleted). The number of up, down, and nonsignificant (ns) genes is shown in the legends. (**B**) Venn diagrams illustrating the number of genes identified

*Figure 8 continued*

as brain EC-enriched (left) or brain EC-depleted (right) versus liver, lung, or kidney ECs (adjusted p-values<0.05, DESeq2 Wald test with Benjamini–Hochberg correction). The 1094 genes enriched in brain ECs compared to each other organ, and the 506 genes depleted in brain ECs compared to each other organ, were used for subsequent analysis of the effects of Wnt activation in the various experimental contexts. (**C–F**) In each plot, the x-axis indicates average $\log_2$(fold change) of gene expression in brain ECs compared to liver, lung, and kidney ECs for the 1094 brain EC-enriched genes and 506 brain EC-depleted genes described in (**B**) with known mouse-human homology. Homologous human gene names are shown. The y-axes indicate differential expression [$\log_2$(fold change)] in Passage 1 CHIR-treated ECs compared to Passage 1 DMSO-treated ECs (**C**), in adult mouse pituitary ECs with stabilized β-catenin (gain-of-function, GOF) compared to controls (***Wang et al., 2019***) (**D**), in adult mouse liver ECs with stabilized β-catenin compared to controls (***Munji et al., 2019***) (**E**), or in cultured adult mouse brain ECs with stabilized β-catenin compared to controls (***Sabbagh and Nathans, 2020***) (**F**). Points are highlighted in blue if concordantly regulated (upregulated in both comparisons or downregulated in both comparisons). The number of concordantly upregulated and concordantly downregulated genes is shown. Genes were identified as upregulated or downregulated based on adjusted p-values<0.05, DESeq2 Wald test with Benjamini–Hochberg correction. (**G**) Venn diagrams illustrating the number of brain EC-enriched genes concordantly upregulated with β-catenin GOF (top) and the number of brain EC-depleted genes concordantly downregulated with β-catenin GOF (bottom) for the four comparisons shown in (**C–F**). Complete results of this analysis are provided in ***Supplementary file 4***.

The online version of this article includes the following figure supplement(s) for figure 8:

**Figure supplement 1.** Concordantly Wnt-regulated central nervous system (CNS) endothelial cell (EC)-associated genes in RNA-seq data of Passage 3 ECs.

**Figure supplement 2.** Pathway analysis of endothelial cells (ECs) with Wnt activation.

pathways that might be differentially regulated by the two strategies for activating Wnt/β-catenin signaling employed in the experiments above: CHIR treatment, which increases β-catenin stability by inhibiting GSK-3, or direct stabilization of β-catenin. We tested lists of upregulated genes in (i) our Passage 1 ECs treated with CHIR versus DMSO, (ii) Passage 3 ECs treated with CHIR versus DMSO, and (iii) pituitary ECs with β-catenin stabilization versus controls (***Wang et al., 2019***), against the Hallmark gene set collection (***Liberzon et al., 2015***; *Figure 8—figure supplement 2*, *Supplementary file 5*). In all three comparisons, the *Wnt/β-catenin signaling* gene set was significantly enriched (***Figure 8—figure supplement 2A***). Similarly, the *Notch signaling*, *TNFα signaling via NF-κB*, *KRAS signaling up*, and several additional gene sets were consistently enriched in all three comparisons (***Figure 8—figure supplement 2A and B***, ***Supplementary file 5***), suggesting similar regulation by GSK-3 inhibition and direct β-catenin stabilization. In contrast, the *PI3K AKT mTOR signaling* gene set was enriched in Passage 1 ECs, but not in Passage 3 ECs or pituitary ECs. Similarly, the gene set *mTORC1 signaling* was enriched in Passage 1 ECs and pituitary ECs, but genes driving this enrichment were distinct (***Figure 8—figure supplement 2C***), and this gene set was not enriched in Passage 3 ECs. Thus, given the known, bidirectional interactions of GSK-3 and AKT/mTOR pathway components (***Hermida et al., 2017***), these results suggest that CHIR-mediated inhibition of GSK-3 may transiently activate this pathway in Passage 1 ECs. Conversely, the gene set *TGF-β signaling* was enriched only in pituitary ECs with β-catenin stabilization (***Figure 8—figure supplement 2***, ***Supplementary file 5***). Taken together, these results, coupled with those of our *CTNNB1* knockdown experiments and gene correlation network analysis, suggest a central role for β-catenin as a key effector of CHIR-mediated signaling, but also highlight some potential differences in the pathways activated in response to CHIR treatment versus β-catenin stabilization. Differences in other aspects of these two experimental paradigms (in vitro versus in vivo, naïve versus CNS-proximal, human versus mouse), however, caution against overinterpretation of these results.

## Discussion

The Wnt/β-catenin signaling pathway plays a central role in CNS angiogenesis and in establishing the unique properties of CNS ECs (***Liebner et al., 2008***; ***Stenman et al., 2008***; ***Daneman et al., 2009***; ***Kuhnert et al., 2010***; ***Cullen et al., 2011***; ***Vanhollebeke et al., 2015***; ***Cho et al., 2017***). In this work, we investigated the role of Wnt/β-catenin signaling in induction of BBB properties in a human EC model using naïve endothelial progenitors derived from hPSCs. We reasoned that these immature EPCs (***Lian et al., 2014***) would be similar to the immature endothelium in the perineural vascular plexus and thus competent to acquire CNS EC phenotypes in response to Wnt activation. To activate Wnt signaling, we evaluated the widely used ligand Wnt3a (***Liebner et al., 2008***) and the GSK-3 inhibitor CHIR.

We found that CHIR treatment robustly induced several canonical CNS EC molecular phenotypes, including a marked induction of GLUT-1, upregulation of claudin-5, and downregulation of PLVAP, which correlated with differential gene expression in RNA-seq data. We also observed a functional decrease in paracellular permeability. Further, using RNA-seq and western blotting, we identified LSR (angulin-1) as CHIR-induced in this system, supporting the notion that this highly CNS EC-enriched tricellular tight junction protein (*Daneman et al., 2010a*; *Sohet et al., 2015*) is Wnt-regulated. In RNA-seq data, we observed differential expression of known CNS EC-enriched/depleted and Wnt-regulated genes including upregulated *LEF1*, *AXIN2*, *APCDD1*, *ABCG2*, *SOX7*, *ZIC3*, *FLVCR2*, *JAM2*, and *RBP1*, and downregulated *PLVAP*, *FABP4*, *SMAD6*, and *SLIT2*. These RNA-seq data should therefore be useful in generating hypotheses of BBB-associated genes regulated by Wnt activation in ECs for future functional studies. Our work also defines an important set of phenotypes for which Wnt activation in ECs is not sufficient in our system: in the context of vesicle trafficking, we observed caveolin-1 (*CAV1*) upregulation, no change in mean functional endocytosis, virtually no expression of *MFSD2A*, and high absolute *PLVAP* abundance in RNA-seq data despite CHIR-mediated downregulation. Given roles of brain pericytes in regulating PLVAP, MFSD2A, and functional transcytosis (*Armulik et al., 2010*; *Daneman et al., 2010b*; *Ben-Zvi et al., 2014*; *Stebbins et al., 2019*), and the observation that MFSD2A is Wnt-regulated in pituitary ECs in vivo (*Wang et al., 2019*), where pericytes are present, it is plausible that pericyte-derived cues are necessary in addition to Wnts to achieve the characteristically low rate of CNS EC pinocytosis. Next, while *ABCG2* (BCRP) was Wnt-induced in our system, other hallmark efflux transporters were not Wnt-regulated and either expressed at low levels (e.g., *ABCC4*, encoding MRP-4) or not expressed (e.g., *ABCB1*, encoding P-glycoprotein). Notably, however, *Abcb1a* was Wnt-regulated in the three other β-catenin stabilization experiments from the literature that we evaluated (*Munji et al., 2019*; *Wang et al., 2019*; *Sabbagh and Nathans, 2020*). Thus, pericyte-derived cues, astrocyte-derived cues, and/or activation of the pregnane X or other nuclear receptors may be important for complete acquisition of the complement of CNS EC efflux transporters (*Bauer et al., 2004*; *Berezowski et al., 2004*; *Praça et al., 2019*).

CHIR is widely used to activate Wnt/β-catenin signaling in cell culture (*Lian et al., 2012*; *Lian et al., 2014*; *Patsch et al., 2015*; *Sakaguchi et al., 2015*; *Gomez et al., 2019*; *Pellegrini et al., 2020*; *Guo et al., 2021*). It remains unknown, however, to what extent CHIR-mediated inhibition of GSK-3 in ECs mimics the effects of Wnt ligand-induced inhibition of GSK-3 or direct stabilization of β-catenin. In our system, although the GLUT-1-inductive effect of CHIR was partially inhibited by β-catenin knockdown and our RNA-seq data revealed a transcriptional response characteristic of canonical Wnt signaling, it is possible that CHIR affects other signaling pathways, as suggested by pathway enrichment analysis. Thus, employing ligand-based strategies to activate Wnt signaling will be an important next step. Our RNA-seq data suggest that the receptors and coreceptors necessary to transduce Wnt7 and Norrin signaling (e.g., *FZD4*, *LRP6*, *RECK*, *ADGRA2* [*GPR124*], *TSPAN12*, *DVL2*) are expressed by hPSC-derived ECs (*Figure 7—figure supplement 6*). Given evidence that Wnt ligands have poor solubility (*Janda et al., 2012*) and our preliminary data suggesting that supplementation of culture medium with Wnt7a and Wnt7b is largely ineffective in activating Wnt/β-catenin signaling in this system, special emphasis should be placed on strategies that present Wnt7a, Wnt7b, and/or Norrin in a manner that concentrates ligands at the cell surface, for example, by using direct cocultures of endogenously Wnt-producing cells (neural progenitors or astrocytes) or Wnt-overexpressing cells. Importantly, neural progenitor cells and astrocytes likely would also contribute other yet-unidentified ligands important for acquisition of CNS EC phenotype. Finally, it would also be informative to directly compare CHIR and/or Wnt ligand treatment to direct stabilization of β-catenin in this system, for example, by generating an hPSC line with inducible expression of a dominant active β-catenin.

We also directly addressed the hypothesis that immature ECs are more plastic, that is, more competent to acquire BBB properties upon Wnt activation than mature ECs. This hypothesis is supported by existing observations that ectopic expression of Wnt7a is sufficient to induce GLUT-1 expression in non-CNS regions of the mouse embryo (*Stenman et al., 2008*), but β-catenin stabilization in adult mouse liver and lung ECs produces only a slight effect (*Munji et al., 2019*). We repeated our CHIR treatment paradigm in hPSC-derived ECs after an extended period of in vitro culture (Passage 4 ECs) and observed much weaker induction of GLUT-1 and no pro-proliferative effect. Thus, our results support this hypothesis and suggest that the loss of BBB developmental plasticity in ECs is an intrinsic, temporally controlled process rather than a result of the peripheral organ environment. The molecular

mechanisms underlying this loss of plasticity remain poorly understood. While previous studies have demonstrated that the level of Wnt/β-catenin signaling in CNS ECs peaks early in development and subsequently declines (*Corada et al., 2019*; *Hübner et al., 2018*), this finding does not address mechanisms underlying the competence of ECs (CNS and non-CNS) to respond to Wnt signals. In RNA-seq data of Passage 3 control (DMSO-treated) ECs, *LEF1* and *TCF7* were strongly downregulated compared to Passage 1 cells. This result suggests that low baseline expression of these transcription factors, which form a complex with nuclear β-catenin to regulate Wnt target genes, may partially explain the poor efficacy of CHIR in matured ECs, although additional work is necessary to assess the functional relevance of these differences. Interestingly, ECs in non-BBB-forming regions of the CNS (i.e., circumventricular organs), and in the anterior pituitary, which is directly proximal to the CNS, retain some of their plasticity in adulthood (*Wang et al., 2019*), possibly as the result of a delicate balance between Wnt ligands and Wnt-inhibitory factors in these regions. Our model should facilitate additional systematic examination of factors that may enhance or attenuate EC Wnt responsiveness.

Finally, our work establishes an improved hPSC-based model for investigating mechanisms of BBB development in naïve ECs. hPSCs are an attractive model system to complement in vivo animal studies because they (i) are human, (ii) permit investigation of developmental processes in contrast to primary or immortalized cells, (iii) are highly scalable, (iv) can be derived from patients to facilitate disease modeling and autologous coculture systems, and (v) are genetically tractable. While widely used hPSC-based BBB models are useful for measuring molecular permeabilities and have been employed to understand genetic contributions to barrier dysfunction (*Vatine et al., 2017*; *Vatine et al., 2019*; *Lim et al., 2017*), they have not been shown to proceed through a definitive endothelial progenitor intermediate (*Lippmann et al., 2012*; *Lu et al., 2021*) and express epithelial-associated genes (*Qian et al., 2017*; *Delsing et al., 2018*; *Vatine et al., 2019*; *Lu et al., 2021*). Thus, new models with developmentally relevant differentiation trajectories and definitive endothelial phenotype are needed for improved understanding of developmental mechanisms. Motivated in part by prior use of ECs derived from hematopoietic progenitors in human cord blood to generate BBB models (*Boyer-Di Ponio et al., 2014*; *Cecchelli et al., 2014*), we and others recently showed that hPSC-derived naïve endothelial progenitors or ECs are good candidates for such a system (*Praça et al., 2019*; *Nishihara et al., 2020*; *Roudnicky et al., 2020a*; *Roudnicky et al., 2020b*). For example, Praça et al. showed that a combination of VEGF, Wnt3a, and retinoic acid directed EPCs to brain capillary-like ECs with moderate TEER similar in order of magnitude to that reported here. We previously showed that BBB-like paracellular barrier characteristics are induced in hPSC-EPC-derived ECs after extended culture in a minimal medium. These so-called EECM-BMEC-like cells had TEER and small molecule permeability similar to primary human brain ECs, well-developed tight junctions, and an immune cell adhesion molecule profile similar to brain ECs in vivo (*Nishihara et al., 2020*). In this study, we showed it was possible to use the small molecule Wnt agonist CHIR to induce additional hallmarks of CNS EC phenotype in hPSC-EPC-derived ECs, including canonical GLUT-1, claudin-5, and PLVAP effects (both Passage 1 and 3 CHIR-treated ECs). However, it is important to note that despite the improvements in CNS EC character with CHIR treatment, further improvements to functional endocytosis, and efflux transporter and solute carrier phenotype should be targets of future study and may be facilitated by cocultures and/ or additional molecular factors. Along these lines, the Passage 1 CHIR-treated CNS-like ECs would be at a differentiation stage well suited to investigate cues subsequent to Wnt signaling that may be key for the induction of additional CNS EC properties. Alternatively, the Passage 3 CHIR-treated CNS-like ECs may be suitable for other BBB modeling applications. In summary, our work has defined the EC response to Wnt activation in a simplified, human system and established a new hPSC-derived in vitro model that will facilitate improved understanding of endothelial barriergenesis.

## Materials and methods

**Key resources table**

| Reagent type (species) or resource | Designation | Source or reference | Identifiers | Additional information |
|---|---|---|---|---|
| Cell line (human) | iPSC: IMR90-4 | Available from WiCell; *Yu et al., 2007* | RRID:CVCL_C437 | |

*Continued on next page*

*Continued*

| Reagent type (species) or resource | Designation | Source or reference | Identifiers | Additional information |
|---|---|---|---|---|
| Cell line (human) | iPSC: WTC11 | Available from Gladstone Institutes; *Kreitzer et al., 2013* | RRID:CVCL_Y803 | |
| Cell line (human) | iPSC: 19-9-11-7TGP-ishcat3 | Laboratory stock | | |
| Cell line (human) | hESC: H9-7TGP-ishcat2 | Laboratory stock *Lian et al., 2013* | | |
| Cell line (human) | hESC: H9-CDH5-eGFP | Laboratory stock *Bao et al., 2017* | | |
| Antibody | Anti-CD31-FITC (mouse monoclonal IgG1, clone AC128) | Miltenyi Biotec | Cat# 130-117-390; RRID:AB_2733637 | |
| Antibody | Anti-CD31-APC (mouse monoclonal IgG1, clone AC128) | Miltenyi Biotec | Cat# 130-119-891; RRID:AB_2784124 | |
| Antibody | Anti-CD34-FITC (mouse monoclonal IgG2a, clone AC136) | Miltenyi Biotec | Cat# 130-113-178; RRID:AB_2726005 | |
| Antibody | Anti-β-catenin-Alexa Fluor 488 (mouse monoclonal IgG1, clone 14) | BD Biosciences | Cat# 562505; RRID:AB_11154224 | (1:100, ICC) |
| Antibody | Anti-GLUT-1 (mouse monoclonal IgG2a, clone SPM498) | Invitrogen | Cat# MA5-11315; RRID:AB_10979643 | (1:100, ICC) (1:500, WB) |
| Antibody | Anti-calponin (mouse monoclonal IgG1, clone hCP) | Sigma-Aldrich | Cat# C2687; RRID:AB_476840 | (1:15,000, ICC) |
| Antibody | Anti-SM22α (rabbit polyclonal) | Abcam | Cat# ab14106; RRID:AB_443021 | (1:1000, ICC) |
| Antibody | Anti-claudin-5 (mouse monoclonal IgG1, clone 4C3C2) | Invitrogen | Cat# 35-2500; RRID:AB_2533200 | (1:100, ICC) (1:500, WB) |
| Antibody | Anti-caveolin-1 (rabbit polyclonal) | Cell Signaling Technology | Cat# 3238; RRID:AB_2072166 | (1:500, ICC) |
| Antibody | Anti-CD31 (rabbit polyclonal) | Lab Vision | Cat# RB-10333-P; RRID:AB_720502 | (1:100, ICC) |
| Antibody | Anti-Ki67 (mouse monoclonal IgG1, clone B56) | BD Biosciences | Cat# 550609; RRID:AB_393778 | (1:100, ICC) |
| Antibody | Anti-VE-cadherin (mouse monoclonal IgG2a, clone BV9) | Santa Cruz Biotechnology | Cat# sc-52751; RRID:AB_628919 | (1:100, ICC) (1:250, WB) |
| Antibody | Anti-β-actin (rabbit monoclonal IgG, clone 13E5) | Cell Signaling Technology | Cat# 4970; RRID:AB_2223172 | (1:1000, WB) |
| Antibody | Anti-PLVAP (rabbit polyclonal) | Prestige Antibodies | Cat# HPA002279; RRID:AB_1079636 | (1:200, ICC) (1:250, WB) |
| Antibody | Anti-LSR (rabbit polyclonal) | Prestige Antibodies | Cat# HPA007270; RRID:AB_1079253 | (1:250, WB) |
| Antibody | Alexa Fluor 488 goat anti-mouse IgG (goat polyclonal) | Invitrogen | Cat# A-11001; RRID:AB_2534069 | (1:200, ICC) |
| Antibody | Alexa Fluor 647 goat anti-rabbit IgG (goat polyclonal) | Invitrogen | Cat# A-21245; RRID:AB_2535813 | (1:200, ICC) |
| Antibody | Alexa Fluor 488 goat anti-mouse IgG1 (goat polyclonal) | Invitrogen | Cat# A-21121; RRID:AB_2535764 | (1:200, ICC) |
| Antibody | Alexa Fluor 647 goat anti-mouse IgG2a (goat polyclonal) | Invitrogen | Cat# A-21241; RRID:AB_2535810 | (1:200, ICC) |
| Antibody | Alexa Fluor 555 goat anti-rabbit IgG (goat polyclonal) | Invitrogen | Cat# A-21428; RRID:AB_2535849 | (1:200, ICC) |

*Continued*

| Reagent type (species) or resource | Designation | Source or reference | Identifiers | Additional information |
|---|---|---|---|---|
| Antibody | IRDye 800CW goat anti-mouse IgG (goat polyclonal) | LI-COR Biosciences | Cat# 926-32210; RRID:AB_621842 | (1:5000, WB) |
| Antibody | IRDye 800CW goat anti-rabbit IgG (goat polyclonal) | LI-COR Biosciences | Cat# 926-32211; RRID:AB_621843 | (1:5000, WB) |
| Antibody | IRDye 680RD goat anti-rabbit IgG (goat polyclonal) | LI-COR Biosciences | Cat# 926-68071; RRID:AB_10956166 | (1:5000, WB) |
| Commercial assay or kit | RNeasy Plus Micro Kit | Qiagen | Cat# 74034 | |
| Chemical compound or drug | CHIR 99021 | Tocris | Cat# 4423 | |
| Chemical compound or drug | Vybrant DyeCycle Green Stain | Invitrogen | Cat# V35004 | |
| Chemical compound or drug | Dextran, Alexa Fluor 488; 10,000 MW, Anionic, Fixable | Invitrogen | Cat# D22910 | |
| Software or algorithm | RSEM | *Li and Dewey, 2011* | RRID:SCR_013027 | v1.3.3 |
| Software or algorithm | Bowtie2 | *Langmead and Salzberg, 2012* | RRID:SCR_016368 | v2.4.2 |
| Software or algorithm | R | R Foundation | RRID:SCR_001905 | v3.6.3 |
| Software or algorithm | DESeq2 | *Love et al., 2014* | RRID:SCR_015687 | v1.26.0 |
| Software or algorithm | biomaRt | *Durinck et al., 2009* | RRID:SCR_019214 | v2.42.1 |
| Software or algorithm | WGCNA | *Langfelder and Horvath, 2008* | RRID:SCR_003302 | v1.70-3 |
| Software or algorithm | Cytoscape | *Shannon et al., 2003* | RRID:SCR_003032 | v3.8.2 |
| Software or algorithm | FIJI/ImageJ | *Schindelin et al., 2012* | RRID:SCR_002285 | v2.0.0-rc-68 |
| Software or algorithm | Image Studio | LI-COR Biosciences | RRID:SCR_015795 | v5.2 |
| Software or algorithm | FlowJo | BD Biosciences | RRID:SCR_008520 | v10.7.1 |
| Software or algorithm | JMP Pro | SAS Institute | RRID:SCR_014242 | v15.0.0 |
| Software or algorithm | Prism | GraphPad Software | RRID:SCR_002798 | v5.0.1 |

## hPSC maintenance

Tissue culture plates were coated with Matrigel, Growth Factor Reduced (Corning, Glendale, AZ). A 2.5 mg aliquot of Matrigel was thawed and resuspended in 30 mL DMEM/F-12 (Life Technologies, Carlsbad, CA), and the resulting solution used to coat plates at 8.7 µg/cm$^2$ (1 mL per well for 6-well plates; 0.5 mL per well for 12-well plates). Plates were incubated at 37°C for at least 1 hr prior to use. hPSCs were maintained on Matrigel-coated plates in E8 medium (STEMCELL Technologies, Vancouver, Canada) at 37°C, 5% $CO_2$. hPSC lines used were IMR90-4 iPSC, WTC11 iPSC, H9-CDH5-eGFP hESC, H9-7TGP-ishcat2 hESC, and 19-9-11-7TGP-ishcat3 iPSC. Medium was changed daily. When hPSC colonies began to touch, typically at approximately 70–80% confluence, cells were passaged using Versene (Life Technologies). Briefly, cells were washed once with Versene, then incubated with Versene for 7 min at 37°C. Versene was removed and cells were dissociated into colonies by gentle spraying with E8 medium. Cells were transferred at a split ratio of 1:12 to a new Matrigel-coated plate containing E8 medium. hPSC cultures were routinely tested for mycoplasma contamination using a PCR-based assay performed by the WiCell Research Institute (Madison, WI).

## Endothelial progenitor cell differentiation

EPCs were differentiated according to previously published protocols (*Lian et al., 2014*; *Bao et al., 2016*; *Nishihara et al., 2020*) with slight modifications. On day –3 (D-3), hPSCs were treated with Accutase (Innovative Cell Technologies, San Diego, CA) for 7 min at 37°C. The resulting single-cell suspension was transferred to 4× volume of DMEM/F-12 (Life Technologies) and centrifuged for 5 min, 200× g. Cell number was quantified using a hemocytometer. Cells were resuspended in E8

medium supplemented with 10 µM ROCK inhibitor Y-27632 dihydrochloride (Tocris, Bristol, UK) and seeded on Matrigel-coated 12-well plates at a density of $(1.5–2.5) \times 10^4$ cells/cm$^2$, 1 mL per well. Cells were maintained at 37°C, 5% CO$_2$. On the following two days (D-2 and D-1), the medium was replaced with E8 medium. The following day (D0), differentiation was initiated by changing the medium to LaSR medium (Advanced DMEM/F-12 [Life Technologies], 2.5 mM GlutaMAX [Life Technologies], and 60 µg/mL L-ascorbic acid 2-phosphate magnesium [Sigma-Aldrich, St. Louis, MO]) supplemented with 7–8 µM CHIR 99021 (Tocris), 2 mL per well. The following day (D1), medium was replaced with LaSR medium supplemented with 7–8 µM CHIR 99021, 2 mL per well. On the following three days (D2, D3, and D4), the medium was replaced with pre-warmed LaSR medium (without CHIR), 2 mL per well.

On D5, EPCs were isolated using CD31 MACS. Cells were treated with Accutase for 15–20 min at 37°C. The resulting cell suspension was passed through a 40 µm cell strainer into an equal volume of DMEM (Life Technologies) supplemented with 10% FBS (Peak Serum, Wellington, CO) and centrifuged for 5 min, 200× g. Cell number was quantified using a hemocytometer. Cells were resuspended in MACS buffer (Dulbecco's phosphate buffered saline without Ca and Mg [DPBS; Life Technologies] supplemented with 0.5% bovine serum albumin [Sigma-Aldrich] and 2 mM EDTA [Sigma-Aldrich]) at a concentration of $10^7$ cells per 100 µL. The CD31-FITC antibody (Miltenyi Biotec, Auburn, CA) was added to the cell suspension at a dilution of 1:50. The cell suspension was incubated for 30 min at room temperature (RT), protected from light. The cell suspension was brought to a volume of 15 mL with MACS buffer and centrifuged for 5 min, 200× g. The supernatant was aspirated and the pellet resuspended in MACS buffer at a concentration of $10^7$ cells per 100 µL. The FITC Selection Cocktail from the EasySep Human FITC Positive Selection Kit (STEMCELL Technologies) was added at a dilution of 1:10, and the cell suspension was incubated for 20 min at RT, protected from light. The Dextran RapidSpheres (magnetic particles) solution from the Selection Kit was added at a dilution of 1:20, and the cell suspension was incubated for an additional 15 min at RT.

The cell suspension was brought to a total volume of 2.5 mL with MACS buffer (for total cell number less than $2 \times 10^8$, the approximate maximum yield from two 12-well plates; for a larger number of plates/cells, a total volume of 5 mL was used). 2.5 mL of cell suspension was transferred to a sterile 5 mL round-bottom flow cytometry tube and placed in the EasySep magnet (STEMCELL Technologies) for 5 min. The magnet was inverted to pour off the supernatant, the flow tube removed, the retained cells resuspended in 2.5 mL of MACS buffer, and the flow tube placed back in the magnet for 5 min. This step was repeated three times, and the resulting cell suspension transferred to a centrifuge tube, and centrifuged for 5 min, 200× g. Cell number was quantified using a hemocytometer. Resulting EPCs were used directly for experiments as described below or cryopreserved in hECSR medium supplemented with 30% FBS and 10% DMSO for later use. hECSR medium is Human endothelial serum-free medium (Life Technologies) supplemented with 1× B-27 supplement (Life Technologies) and 20 ng/mL FGF2 (Waisman Biomanufacturing, Madison, WI).

## Endothelial cell culture and treatment

Collagen IV (Sigma-Aldrich) was dissolved in 0.5 mg/mL acetic acid to a final concentration of 1 mg/mL. Collagen IV-coated plates were prepared by diluting a volume of this stock solution 1:100 in water, adding the resulting solution to tissue culture plates, or #1.5 glass-bottom plates (Cellvis, Sunnyvale, CA) for cells intended for confocal imaging (1 mL per well for 6-well plates, 0.5 mL per well for 12-well plates, 0.25 mL per well for 24-well plates), and incubating the plates for 1 hr at RT. Collagen IV coating solution was removed, and EPCs obtained as described above were suspended in hECSR medium and plated at approximately $3 \times 10^4$ cells/cm$^2$. In some experiments, ligands and small molecules were added to hECSR medium: CHIR 99021 (Tocris) was used at 4 µM except where indicated; DMSO (Sigma-Aldrich) was used as a vehicle control for CHIR; Wnt3a (R&D Systems) was used at 20 ng/mL; doxycycline was used at 1, 2, or 4 µg/mL. The hECSR medium, including any ligands or small molecules, was replaced every other day until confluent (typically 6 days). We denote this time point as 'Passage 1.'

For extended culture, ECs were selectively dissociated and replated as previously described (*Nishihara et al., 2020*). Cells were incubated with Accutase until ECs appeared round, typically 2–3 min at 37°C. The plate was tapped to release the ECs while SMLCs remained attached, and the EC-enriched cell suspension transferred to 4× volume of DMEM/F-12 and centrifuged for 5 min, 200× g. Cells were resuspended in hECSR medium and seeded on a new collagen IV-coated plate at approximately

$3 \times 10^4$ cells/cm$^2$. hECSR medium was replaced every other day until confluent (typically 6 days). The selective dissociation and seeding described above was repeated, and hECSR medium was again replaced every other day until confluent (typically 6 days). We denote this time point as 'Passage 3.' In one experiment, these steps were repeated for another two passages. Except where indicated, CHIR 99021 or vehicle (DMSO) was included in the hECSR medium for the entire duration of culture.

## RNA-seq

RNA-seq was performed on ECs and SMLCs from the IMR90-4 hPSC line. Four independent differentiations were performed, with DMSO- and CHIR-treated ECs at Passage 1 analyzed from all four differentiations. DMSO- and CHIR-treated ECs at Passage 3 were analyzed from three of the four differentiations. DMSO-treated SMLCs at Passage 1 were analyzed from two of the four differentiations. FACS was used to isolate CD31$^+$ ECs and CD31$^-$ SMLCs from mixed Passage 1 cultures. Cells were incubated with Accutase for 10 min at 37°C, passed through 40 μm cell strainers into 4× volume of DMEM/F-12, and centrifuged for 5 min, 200× g. Cells were resuspended in MACS buffer and incubated with CD31-APC antibody (Miltenyi Biotec) for 30 min at 4°C, protected from light. The cell suspension was brought to a volume of 15 mL with MACS buffer and centrifuged at 4°C for 5 min, 200× g. Cells were resuspended in MACS buffer containing 2 μg/mL 4',6-diamidino-2-phenylindole (DAPI; Life Technologies). A BD FACSAria III Cell Sorter (BD Biosciences, San Jose, CA) was used to isolate DAPI$^-$CD31$^+$ cells (live ECs) and DAPI$^-$CD31$^-$ cells (live SMLCs). The resulting cell suspensions were centrifuged at 4°C for 5 min, 200× g, and cell pellets immediately processed for RNA extraction as described below.

RNA was isolated using the RNeasy Plus Micro Kit (Qiagen, Germantown, MD). Buffer RLT Plus supplemented with 1% β-mercaptoethanol was used to lyse cells (pellets from FACS of Passage 1 cells, or directly on plates for Passage 3 ECs). Lysates were passed through gDNA Eliminator spin columns, loaded onto RNeasy MinElute spin columns, washed with provided buffers according to the manufacturer's instructions, and eluted with RNase-free water. Sample concentrations were determined using a NanoDrop spectrophotometer (Thermo Scientific, Waltham, MA) and RNA quality assayed using an Agilent 2100 Bioanalyzer with Agilent RNA 6000 Pico Kit (Agilent, Santa Clara, CA). First-strand cDNA synthesis was performed using the SMART-Seq v4 Ultra Low Input RNA kit (Takara Bio, Mountain View, CA) with 5 ng input RNA followed by nine cycles of PCR amplification and library preparation using the Nextera XT DNA Library Prep Kit (Illumina, San Diego, CA). Sequencing was performed on a NovaSeq 6000 (Illumina), with approximately 40–60 million 150 bp paired-end reads obtained for each sample.

FASTQ files were aligned to the human genome (hg38) and transcript abundances quantified using RSEM (v1.3.3) (*Li and Dewey, 2011*) calling bowtie2 (v2.4.2) (*Langmead and Salzberg, 2012*). Estimated counts from RSEM were input to DESeq2 (v1.26.0) (*Love et al., 2014*) implemented in R (v3.6.3) for differential expression analysis. Elsewhere, transcript abundances are presented as TPM. Differentiation pairing as described above was included in the DESeq2 designs. The Wald test with Benjamini–Hochberg correction was used to generate adjusted p-values. Principal component analysis was performed on counts after the DESeq2 variance stabilizing transformation. Transcription factor annotations were based on the list available at http://humantfs.ccbr.utoronto.ca/ (*Lambert et al., 2018*); secreted and transmembrane annotations were based on the UniProt database (*UniProt Consortium, 2021*). WGCNA (v1.70–3) (*Zhang and Horvath, 2005*; *Langfelder and Horvath, 2008*) was performed on the 14 EC datasets. Genes with an average of fewer than 50 estimated counts across these datasets were excluded, and the DESeq2 variance stabilizing transformation was used to generate the expression matrix for input to WGCNA. The topological overlap matrix (TOM) was constructed using the signed network type and a power of 20. Hierarchical clustering was performed on dissimilarity (1 − TOM) with average linkage. Gene modules were detected by a constant height (0.99) cut of the hierarchical clustering dendrogram with a minimum module size of 30 genes. Module eigengenes (the first principal component of the expression matrix for genes in each module) were computed as described, and the Pearson correlation between module eigengenes and experimental variables (CHIR vs. DMSO: CHIR = 1, DMSO = 0; Passage 3 vs. Passage 1: Passage 3 = 1, Passage 1 = 0) was used to identify modules of interest. Cytoscape (v3.8.3) (*Shannon et al., 2003*) was used to visualize the 30 genes in the green module (strong positive correlation with CHIR treatment) with

the highest intramodular connectivity. The list of genes, corresponding modules, and correlations to experimental variables and module eigengenes is provided in *Supplementary file 3*.

Bulk RNA-seq data from the literature (FASTQ files; see 'Previously published datasets used') were obtained from the Gene Expression Omnibus (GEO). These FASTQ files were aligned to the mouse genome (mm10) and transcript abundances quantified as described above. DESeq2 was used for differential expression analysis as described above. For direct comparison of human and mouse data, the biomaRt package (v2.42.1) (*Durinck et al., 2009*) and Ensembl database (*Yates et al., 2020*) were used to map human gene names to mouse homologs. Venn diagrams were generated using the tool available at http://bioinformatics.psb.ugent.be/webtools/Venn/. To identify solute carrier and efflux transporter genes highly expressed at the human BBB in vivo, we used five human brain scRNA-seq datasets (*Han et al., 2020*; *Hodge et al., 2019*; *La Manno et al., 2016*; *Polioudakis et al., 2019*; *Zhong et al., 2020*; see 'Previously published datasets used') integrated in a previous meta-analysis (*Gastfriend et al., 2021*). *SLC* and *ABC* genes with average expression greater than 100 TPM in ECs across the five independent datasets were selected. For pathway enrichment analysis, lists of upregulated genes ($\log_2$(fold change) > 0, adjusted p-value<0.05, DESeq2 Wald test with Benjamini–Hochberg correction) were tested against the Hallmark gene sets collection (*Liberzon et al., 2015*) using the tool available at http://www.gsea-msigdb.org/gsea/msigdb/annotate.jsp.

## Immunocytochemistry

Immunocytochemistry was performed in 24-well plates. Cells were washed once with 500 µL DPBS and fixed with 500 µL cold (–20°C) methanol for 5 min, except cells intended for calponin/SM22a and CD31/Ki67 detection, which were fixed with 500 µL of 4% paraformaldehyde for 15 min. Cells were washed three times with 500 µL DPBS and blocked in 150 µL DPBS supplemented with 10% goat serum (Life Technologies) for 1 hr at RT, except cells intended for calponin/SM22α detection, which were blocked and permeabilized in DPBS supplemented with 3% BSA and 0.1% Triton X-100, or cells intended for CD31/Ki67 detection, which were blocked and permeabilized in DPBS supplemented with 5% non-fat dry milk and 0.4% Triton X-100. Primary antibodies diluted in 150 µL of the above blocking solutions (see Key resources table for antibody information) were added to cells and incubated overnight at 4°C on a rocking platform. Cells were washed three times with 500 µL DPBS. Secondary antibodies diluted in 150 µL of the above blocking solutions (see Key resources table for antibody information) were added to cells and incubated for 1 hr at RT on a rocking platform, protected from light. Cells were washed three times with 500 µL DPBS, followed by 5 min incubation with 500 µL DPBS plus 4 µM Hoechst 33342 (Life Technologies). Images were acquired using an Eclipse Ti2-E epifluorescence microscope (Nikon, Tokyo, Japan) with a 20× or 30× objective or an A1R-Si+ confocal microscope (Nikon) with a 100× oil objective. Confocal images were acquired with 1 µm slice spacing.

Images were analyzed using FIJI (ImageJ) software. For epifluorescence images, five fields (20× or 30×) were analyzed per well, with 3–4 wells per treatment condition. For quantification of cell number, EC colonies were manually outlined, and the Analyze Particles function was used to estimate the number of nuclei within the EC colonies. Nuclei outside the EC colonies were manually counted. EC purity (% EC) was calculated as the number of nuclei within EC colonies relative to total nuclei. To estimate % GLUT-1$^+$ ECs, cells within the EC colonies with membrane-localized GLUT-1 immunoreactivity were manually counted. To estimate % Ki67$^+$ ECs, cells within the EC colonies with at least one nuclear-localized Ki67 punctum were manually counted. For quantification of fluorescence intensity in epifluorescence images, EC colonies were manually outlined, and the Measure function was used to obtain the mean fluorescence intensity for each image channel (fluorophore). A cell-free area of the plate was similarly quantified for background subtraction. Following background subtraction, the mean fluorescence intensity of each protein of interest was normalized to the mean fluorescence intensity of Hoechst to correct for effects of cell density. For confocal images, 3–4 fields (100×) containing only VE-cadherin$^+$ ECs were analyzed per well, with four wells per treatment condition. The first slice with visible nuclei (closest to glass) was defined as $Z = 0$, and the Measure function was used to obtain the mean fluorescence intensity for each image channel (fluorophore) in each slice from $Z = 0$ to $Z = 7$ µm. A cell-free area of the plate was similarly quantified for background subtraction. After background subtraction, to approximate total abundance (area under the fluorescence versus $Z$ curve

[AUC]) for each channel, mean fluorescence intensities were summed across all slices. AUCs for the proteins of interest were normalized to Hoechst AUC.

## Cell cycle analysis

Passage 1 cultures were dissociated by treatment with Accutase for 10 min at 37°C. Cell suspensions were passed through 40 µm cell strainers into 4× volume of DMEM/F-12 and centrifuged for 5 min, 200× g. Approximately $5 \times 10^5$ cells per replicate were resuspended in MACS buffer and incubated with the CD31-APC antibody (Miltenyi Biotec) for 30 min at 4°C, protected from light. Cell suspensions were brought to a volume of 5 mL with MACS buffer and centrifuged at 4°C for 5 min, 200× g. Cells were resuspended in 500 µL MACS buffer containing 2 µg/mL DAPI and 0.5 µL Vybrant DyeCycle Green Stain (Invitrogen) and incubated at RT for 1 hr, protected from light. Cells were analyzed on an Attune NxT flow cytometer (Invitrogen). FlowJo software (BD Biosciences) was used to gate CD31[+] cells and quantify the percentage of S/G2/M phase cells.

## Western blotting

To enrich samples from Passage 1 cultures for ECs, the Accutase-based selective dissociation method described above was employed. Dissociated cells were centrifuged for 5 min, 200× g, and resulting cell pellets were lysed in RIPA buffer (Rockland Immunochemicals, Pottstown, PA) supplemented with 1× Halt Protease Inhibitor Cocktail (Thermo Scientific). Passage 3 cells were lysed with the above buffer directly on plates. Lysates were centrifuged at 4°C for 5 min, 14,000× g, and protein concentration in supernatants quantified using the Pierce BCA Protein Assay Kit (Thermo Scientific). Equal amounts of protein were diluted to equal volume with water, mixed with sample buffer, and heated at 95°C for 5 min, except lysates intended for GLUT-1 western blotting, which were not heated. Samples were resolved on 4–12% Tris-Glycine gels and transferred to nitrocellulose membranes. Membranes were blocked for 1 hr in Tris-buffered saline plus 0.1% Tween-20 (TBST) supplemented with 5% non-fat dry milk. Primary antibodies (see Key resources table for antibody information) diluted in TBST plus 5% non-fat dry milk were added to membranes and incubated overnight at 4°C on a rocking platform. Membranes were washed five times with TBST. Secondary antibodies (see Key resources table for antibody information) diluted in TBST were added to membranes and incubated for 1 hr at RT on a rocking platform, protected from light. Membranes were washed five times with TBST and imaged using an Odyssey 9120 (LI-COR, Lincoln, NE). Band intensities were quantified using Image Studio software (LI-COR).

## Dextran accumulation assay

A fixable, Alexa Fluor 488-conjugated dextran with an average molecular weight of 10 kDa (Invitrogen) was used as a tracer to estimate total fluid-phase endocytosis. Dextran was added at 10 µM to the medium of Passage 1 cultures. Plates were incubated on rotating platforms at 37 or 4°C for 2 hr. For inhibitor experiments, 20 µM chlorpromazine (Sigma), 100 U/mL nystatin (Sigma), or 2 µM rottlerin (Tocris) were added to the medium 30 min prior to addition of dextran. Medium was removed and cells were washed once with DPBS, and then incubated with Accutase for 10 min at 37°C. Cell suspensions were passed through 40 µm cell strainers into 4× volume of DMEM/F-12 and centrifuged for 5 min, 200× g. Cells were resuspended in MACS buffer and incubated with the CD31-APC antibody (Miltenyi Biotec) for 30 min at 4°C, protected from light. Cell suspensions were brought to a volume of 5 mL with MACS buffer and centrifuged at 4°C for 5 min, 200× g. Pellets were resuspended in DPBS supplemented with 4% paraformaldehyde and incubated for 15 min at RT, protected from light. Cells were centrifuged for 5 min, 200× g. Pellets were resuspended in MACS buffer and analyzed on a BD FACSCalibur flow cytometer (BD Biosciences). FlowJo software was used to gate CD31[+] cells and quantify geometric mean fluorescence intensity and CV of dextran. For imaging, the dextran accumulation assay was performed on cells cultured on #1.5 glass-bottom plates. After 2 hr of dextran treatment, medium was removed and cells washed with DPBS. Cells were fixed with 4% paraformaldehyde for 15 min. Cells were washed three times with 500 µL DPBS and blocked and permeabilized with DPBS supplemented with 10% goat serum and 0.1% Triton X-100 for 1 hr at RT. Cells were stained with the caveolin-1 primary antibody and imaged on a confocal microscope as described above.

## Transendothelial electrical resistance and sodium fluorescein permeability

Transwell inserts (6.5 mm diameter with 0.4 µm pore polyester filters) (Corning) were coated with 50 µL of a solution of collagen IV (400 µg/mL) and fibronectin (100 µg/mL) in water for 4 hr at 37°C. Passage 3 DMSO- and CHIR-treated ECs were seeded on Transwell inserts at $10^5$ cells/cm$^2$ in hECSR medium supplemented with DMSO or CHIR. Medium volumes were 200 µL for the apical chamber and 800 µL for the basolateral chamber. Beginning the day after seeding, TEER was measured daily for 6 days using an EVOM2 epithelial voltohmmeter with STX2 chopstick electrodes (World Precision Instruments, Sarasota, FL). Medium was replaced every other day. TEER values were corrected by subtracting the resistance of a collagen IV/fibronectin-coated Transwell insert without cells and multiplying by the filter surface area of 0.33 cm$^2$. Permeability of endothelial monolayers to sodium fluorescein was assessed 6 days after seeding cells on Transwell inserts. Medium in both apical and basolateral chambers was replaced and cells returned to the incubator for 1 hr. Medium in apical chambers, including the apical chamber of a collagen IV/fibronectin-coated Transwell insert without cells, was then replaced with medium supplemented with 10 µM sodium fluorescein (Sigma-Aldrich), and plates placed on an orbital platform in an incubator. At 15, 30, 45, and 60 min, an 80 µL sample of the basolateral chamber medium was withdrawn from each Transwell, transferred to a 96-well plate, and 80 µL fresh medium replaced in the basolateral chamber of each Transwell. At 60 min, an 80 µL sample of apical chamber medium was also withdrawn from each Transwell and transferred to the 96-well plate. 80 µL of medium lacking sodium fluorescein was also transferred to the 96-well plate for background subtraction. Fluorescence intensity of all samples was measured using an Infinite M1000 PRO plate reader (Tecan, Männedorf, Switzerland) with 485 nm excitation and 530 nm emission wavelengths. Background-subtracted fluorescence intensity values at the 30, 45, and 60 min timepoints were corrected for sampling-induced dilution as previously described (*Stebbins et al., 2016*). The endothelial permeability coefficient ($P_e$), which is a concentration-independent parameter corrected for the permeability of a cell-free Transwell insert, was calculated as previously described (*Stebbins et al., 2016*).

## Statistics

Individual wells of cultured cells that underwent identical experimental treatments are defined as replicates, and all key experiments were repeated using multiple independent hPSC differentiations. Detailed information about replication strategy is provided in figure legends. Student's *t* test was used for comparison of means from two experimental groups. One-way analysis of variance (ANOVA) was used for comparison of means from three or more experimental groups, followed by Dunnett's post-hoc test for comparison of multiple treatments to a single control, or Tukey's honest significant difference (HSD) post-hoc test for multiple pairwise comparisons. When data from multiple differentiations were combined, two-way ANOVA (one factor being the experimental treatment and one factor being the differentiation) was used for comparison of means to achieve blocking of differentiation-based variability, followed by post-hoc tests as described above if more than two experimental treatments were compared. For fluorescence intensities (a.u.), two-way ANOVA was performed prior to normalization of these values to the control group within each differentiation (for visualization in plots). Statistical tests were performed in JMP Pro (v15.0.0). For RNA-seq differential expression analysis, the DESeq2 Wald test with Benjamini–Hochberg correction was used to calculate p-values. Descriptions of the statistical tests used are provided in figure legends.

## Acknowledgements

We acknowledge the University of Wisconsin–Madison Biotechnology Center Gene Expression Center and DNA Sequencing Facility for providing library preparation and next-generation sequencing services. We acknowledge the University of Wisconsin–Madison Biochemistry Optical Core for use of a confocal microscope. We acknowledge the University of Wisconsin Carbone Cancer Center Flow Cytometry Laboratory (supported by NIH Cancer Center Support Grant P30 CA014520) for use of a flow cytometer and for performing FACS. We thank Richard Daneman for advice related to this work.

## Additional information

### Competing interests

Benjamin D Gastfriend, Hideaki Nishihara, Britta Engelhardt, Sean P Palecek, Eric V Shusta: Inventor on a provisional US patent application (63/185815) related to this work. The other authors declare that no competing interests exist.

### Funding

| Funder | Grant reference number | Author |
| --- | --- | --- |
| National Institutes of Health | R01 NS103844 | Sean P Palecek<br>Eric V Shusta |
| National Institutes of Health | R01 NS107461 | Sean P Palecek<br>Eric V Shusta |
| National Institutes of Health | T32 GM008349 | Benjamin D Gastfriend |
| National Science Foundation | 1747503 | Benjamin D Gastfriend |
| Schweizerischer Nationalfonds zur Förderung der Wissenschaftlichen Forschung | 310030_189080 | Britta Engelhardt |
| Bern Center for Precision Medicine | | Britta Engelhardt |
| Japan Society for the Promotion of Science | Overseas Research Fellowship | Hideaki Nishihara |

The funders had no role in study design, data collection and interpretation, or the decision to submit the work for publication.

### Author contributions

Benjamin D Gastfriend, Conceptualization, Formal analysis, Investigation, Methodology, Writing – original draft; Hideaki Nishihara, Conceptualization, Investigation, Methodology, Validation, Writing – review and editing; Scott G Canfield, Conceptualization, Methodology, Writing – review and editing; Koji L Foreman, Conceptualization, Investigation, Writing – review and editing; Britta Engelhardt, Sean P Palecek, Eric V Shusta, Conceptualization, Funding acquisition, Supervision, Writing – review and editing

### Author ORCIDs

Benjamin D Gastfriend http://orcid.org/0000-0002-4677-1455
Sean P Palecek http://orcid.org/0000-0003-4917-5584
Eric V Shusta http://orcid.org/0000-0002-4297-0158

### Decision letter and Author response

Decision letter https://doi.org/10.7554/eLife.70992.sa1
Author response https://doi.org/10.7554/eLife.70992.sa2

## Additional files

### Supplementary files

• Transparent reporting form

• Supplementary file 1. RNA-sequencing gene expression data for human pluripotent stem cell (hPSC)-derived endothelial cells (ECs) and smooth muscle-like cells (SMLCs). Abundances are provided in transcripts per million (TPM).

• Supplementary file 2. RNA-sequencing differential expression analysis of human pluripotent stem cell (hPSC)-derived endothelial cells (ECs). (A–C) DESeq2-derived average expression (baseMean),

log$_2$(fold change), Wald statistic, p-value (Wald test), and adjusted p-value (Benjamini–Hochberg correction) are shown. (A) Passage 1 CHIR-treated ECs versus Passage 1 DMSO-treated ECs. (B) Passage 3 CHIR-treated ECs versus Passage 3 DMSO-treated ECs. (C) Passage 3 DMSO-treated ECs versus Passage 1 DMSO-treated ECs. (D) Lists of upregulated and downregulated genes comprising the intersection of the comparisons in (A) and (B), used to generate Venn diagram in *Figure 7E*.

• Supplementary file 3. Gene correlation network analysis. For each gene, module assignment, correlation with experimental variables (CHIR treatment and passage number), and module membership (correlation between gene and module eigengene) are shown. Genes with an average of fewer than 50 estimated counts were excluded from this analysis.

• Supplementary file 4. Wnt-regulated endothelial cell (EC) genes in multiple contexts. (A) Differential expression analysis of P7 murine brain, liver, lung, and kidney ECs (*Sabbagh et al., 2018*). DESeq2-derived average expression (baseMean), log$_2$(fold change), Wald statistic, p-value (Wald test), and adjusted p-value (Benjamini–Hochberg correction) are shown. (B) Lists of brain-enriched and brain-depleted genes comprising the intersection of the comparisons in (A). (C–E) Differential expression analysis of adult murine ECs with β-catenin-stabilization versus controls from pituitary (*Wang et al., 2019*) (C), liver (*Munji et al., 2019*) (D), and brain ECs cultured in vitro (*Sabbagh and Nathans, 2020*) (E). DESeq2-derived average expression (baseMean), log$_2$(fold change), Wald statistic, p-value (Wald test), and adjusted p-value (Benjamini–Hochberg correction) are shown. (F) Lists of concordantly Wnt-regulated genes in Passage 1 human pluripotent stem cell (hPSC)-derived ECs and the three comparisons shown in (C–E), from the set of brain-enriched and brain-depleted genes identified in (B).

• Supplementary file 5. Pathway analysis of endothelial cell (ECs) with Wnt activation. (A–C) Enriched gene sets from the Hallmark gene set collection are shown. For each gene set, the total number of genes in the gene set (K), the number of genes enriched in a given comparison (k), the enrichment ratio (k/K), and the false discovery rate (FDR) are shown. (A) Passage 1 CHIR-treated ECs versus Passage 1 DMSO-treated ECs. (B) Passage 3 CHIR-treated ECs versus Passage 3 DMSO-treated ECs. (C) Adult mouse pituitary ECs with β-catenin-stabilization versus controls (*Wang et al., 2019*).

## Data availability

RNA-seq data have been deposited in GEO under accession number GSE173206.

The following dataset was generated:

| Author(s) | Year | Dataset title | Dataset URL | Database and Identifier |
|---|---|---|---|---|
| Gastfriend BD, Palecek SP, Shusta EV | 2021 | RNA-sequencing of human pluripotent stem-cell derived endothelial cells under control and Wnt-activating conditions | https://www.ncbi.nlm.nih.gov/geo/query/acc.cgi?acc=GSE173206 | NCBI Gene Expression Omnibus, GSE173206 |

The following previously published datasets were used:

| Author(s) | Year | Dataset title | Dataset URL | Database and Identifier |
|---|---|---|---|---|
| Sabbagh MF, Heng J, Luo C, Castanon RG, Nery JR, Rattner A, Goff LA, Ecker JR, Nathans J | 2018 | Transcriptional and Epigenomic Landscapes of CNS and non-CNS Vascular Endothelial Cells | https://www.ncbi.nlm.nih.gov/geo/query/acc.cgi?acc=GSE111839 | NCBI Gene Expression Omnibus, GSM3040844 GSM3040845 GSM3040852 GSM3040853 GSM3040858 GSM3040859 GSM3040864 GSM3040865 |
| Munji R, Daneman R | 2018 | Gene expression profiles of liver and lung endothelial cells during normal and upregulation of Wnt/beta-catenin signaling | https://www.ncbi.nlm.nih.gov/geo/query/acc.cgi?acc=GSE95201 | NCBI Gene Expression Omnibus, GSM2498580 GSM2498581 GSM2498582 GSM2498583 GSM2498584 GSM2498585 GSM2498586 GSM2498587 |

*Continued on next page*

*Continued*

| Author(s) | Year | Dataset title | Dataset URL | Database and Identifier |
|---|---|---|---|---|
| Wang Y, Sabbagh MF, Gu X, Rattner A, Williams J, Nathans J | 2019 | The role of beta-catenin signaling in regulating barrier vs. non-barrier gene expression programs in circumventricular organ and ocular vasculatures | https://www.ncbi.nlm.nih.gov/geo/query/acc.cgi?acc=GSE122117 | NCBI Gene Expression Omnibus, GSM3455653 GSM3455654 GSM3455657 GSM3455658 GSM3455661 GSM3455662 GSM3455665 SM3455666 |
| Sabbagh MF, Nathans J | 2020 | A genome-wide view of the de-differentiation of central nervous system endothelial cells in culture | https://www.ncbi.nlm.nih.gov/geo/query/acc.cgi?acc=GSE118731 | NCBI Gene Expression Omnibus, GSM4160534 GSM4160535 GSM4160536 GSM4160537 GSM4160538 GSM4160539 GSM4160540 GSM4160541 GSM4160542 GSM4160543 |
| Allen Institute | 2019 | Human Multiple Cortical Areas SMART-seq | https://portal.brain-map.org/atlases-and-data/rnaseq/human-multiple-cortical-areas-smart-seq | Allen Brain Map, human-multiple-cortical-areas-smart-seq |
| Polioudakis D | 2019 | Cortical Development Expression | http://solo.bmap.ucla.edu/shiny/webapp/ | UCLA, CoDex |
| Guo G, Han X, Zhou Z, Fei L, Sun H, Wang R, Wang J, Chen H | 2020 | Construction of A Human Cell Landscape by Single-cell mRNA-seq | https://www.ncbi.nlm.nih.gov/geo/query/acc.cgi?acc=GSE134355 | NCBI Gene Expression Omnibus, GSM3980129 GSM4008656 GSM4008657 GSM4008658 |
| LaManno G, Gyllborg D, Arenas E, Linnarsson S | 2016 | Single Cell RNA-seq Study of Midbrain and Dopaminergic Neuron Development in Mouse, Human, and Stem Cells | https://www.ncbi.nlm.nih.gov/geo/query/acc.cgi?acc=GSE76381 | NCBI Gene Expression Omnibus, GSE76381 |
| Zhong S, Wu Q, Wang X | 2019 | Characterization of the transcriptional landscape of human developing hippocampus by scRNA-seq | https://www.ncbi.nlm.nih.gov/geo/query/acc.cgi?acc=GSE119212 | NCBI Gene Expression Omnibus, GSE119212 |

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
