## [Editor Report]

This work provides a considerable novelty to the view on barrier induction in endothelial cells in the central nervous system by the Wnt/β-catenin pathway. By recapitulating the endothelial differentiations events towards a blood–brain barrier phenotype, the presented results offer novel ways to establish cell culture systems that faithfully recapitulate the blood–brain barrier ex vivo, and thereby supporting basic as well as translational research.

---

## [Decision Letter]

**Decision letter after peer review:**

Thank you for submitting your article "Wnt signaling mediates acquisition of blood-brain barrier properties in naïve endothelium derived from human pluripotent stem cells" for consideration by *eLife*. Your article has been reviewed by 4 peer reviewers, one of whom is a member of our Board of Reviewing Editors, and the evaluation has been overseen by Didier Stainier as the Senior Editor. The following individual involved in review of your submission has agreed to reveal their identity: Maxime Culot (Reviewer #3).

Essential revisions:

1) The authors need to explore in more depth the differences in the expression profile between naïve endothelial progenitors and (partially) differentiated ECs by RNA-Seq.

2) Moreover, the authors should investigate other pathways induced by CHIR (GSK3 substrates; Akt, IκB-α, presenilin etc.) in comparison with dominant-active β-catenin.

3) Given that the mechanistic insight in the discrepancies between the stimulations with CHIR and Wnt7a/b, as well as with conditioned medium from neurospheres and astrocytes is limited, the authors should omit these data and add this approach to the discussion.

4) The authors should clearly delineate the aim of the study that, as perceived by the reviewers, appears to be the characterization of the Wnt pathway effects on directing human pluripotent stem cells toward the differentiation to a BBB endothelium.

*Reviewer #1 (Recommendations for the authors):*

See public review.

*Reviewer #2 (Recommendations for the authors):*

While this paper may be a worthwhile method development paper (as published in PLoS One, Science Reports or Microvascular research; eg. Cecchelli 2014 doi:10.1371/journal.pone.0099733; Linville 2020 doi: 10.1016/j.mvr.2020.104042), it lacks scientific novelty and depth required for a publication in *eLife*.

Numerous papers have identified Wnt3a for mesoderm induction and Wnt7a as the major brain EC ligand, and the use of a chemical GSK-3 inhibitor for more powerful Wnt signaling pathway activation is not novel or surprising. Indeed many other models and Wnt-pathway response papers have been published, numerous in the last two years (e.g. Laksitorini et al., Sci Rep 2019 9(1):19718. "Modulation of Wnt/β-catenin signaling promotes blood-brain barrier phenotype in cultured brain endothelial cells").

In addition one major claim of archievement is the analysis of EC maturity linked to Wnt-responsiveness. However, it has already been shown that there is a crucial timing difference in the responsiveness to Wnt signals both in vivo and in vitro (including, but not at all limited to the following examples: Hübner et al., Nat commun 2018 Nov 19;9(1):4860 "Wnt/β-catenin signaling regulates VE-cadherin-mediated anastomosis of brain capillaries by counteracting S1pr1 signaling"; Corada et al., Circ Res 2019 124(4):511-525 "Fine-Tuning of Sox17 and Canonical Wnt Coordinates the Permeability Properties of the Blood-Brain Barrier").

Additionally many of the mentioned examples not only analyze the contribution of WNT signaling, but also link other signaling pathways, whereas this paper identifies other but unknown signaling contributions for establishing ECs with full BBB properties.

Given the lack of novelty and the following mentioned difficulties of biological sample variations, I do not recommend the paper for publication in *eLife*.

Other comments:

Figure 1 shows more than factor2 differences between different differentiation samples (in terms of glut-1 positive cells), making this a less robust model than indicated.

The overall percentage of Glut-1 positive cells in Figure 1 and 2 is very low, making the comparison between the effectiveness of the different ligands while statistically significant compared to the control, most likely not significant compared between the ligands. Therefore the claims of e.g. Wnt7a being more effective is based on one of 3 samples of the same differentiation. All these comaparative claims will have to be either supported by much more data or taken out. The only clear result is that CHIR can stimulate expression of Glut-1 in around 90% of the cells, whereas the offered Wnt ligands affect 2-4%. To me this suggests that soluble WNT ligands in the culture medium are least effective in stimulating a response. It would have been most interesting to explore if co-cultures and hence the supply in e.g. Cytonems or other vesicle or membrane-bound delivery systems would make a huge difference. However this is not the scope of the paper.The analysis of mean fluorescence intensity of claudin-5, caveolin-1, and GLUT-1 normalized to Hoechst mean fluorescence intensity within the area of claudin-5+ ECs only is less informative, as one highly expressing cell will outweigh many low expressing cells and there is no published link of an endothelial cell resembling a BBB endothelial cells more closely in correlation with their Glut-1 expression intensity.

Unfortunately, when analysing the effects of conditioned media in Figure3 only those relative quantifications were used and again the spread between the two different differentiation rounds was so high, that a comparative analysis is hindered and not represented by statistical analysis compared to control only. For example the Caveolin/Hoest values of the orange dataset were increased in the CHIR treatment, but unchanged in Control and NR-CM, but in contrast in the green dataset the NR-CM resulted in a higher index compared to CHIR or the control. Again were no statistical comparison was done between samples, the drawn conclusions can not indicate there are any.

Figure 5 Anal;yses the effect of passage number in combination with CHIR treatment. However the individual differentiation samples again behave partially opposite: the green sample Glut1/ß-actin is only weakle elevated in passage 1 and upregulated after passage3, the purple sample is high in passage 1 and becomes downregulated after passage 3.

Figure 7 the effects in the mature EC should be directly compared to the effects in more naïve from the same differentiation.

*Reviewer #3 (Recommendations for the authors):*

Although I have a positive opinion on the manuscript, I personally think that the manuscript could be slightly simplified (and shorten). Therefore, I have listed below several points that I think should be addressed (and possibly used to shorten the manuscript) by the authors before publication of the manuscript.

– Page 8 Line 173- 192 : "In the CNS, neural progenitors ….. ….. the Wnt ligand signal (analyzed further below)”.

This part and associated figure 3 compare the effects of neural rosette-conditioned medium and astrocyte-conditioned medium to the one of WNT7A and CHIR based on the fact that both cell types were suggested to express WNT7A (Vatine et al., 2016; Shang et al., 2018)..

Although the data are interesting per se., these data in my opinion, do not really have an added value to the main topic of the study (i.e deciphering the effects of wnt signaling in human endothelial cell progenitor with a focus on barrier formation).

In addition, the reported effects of neural rosette and astrocyte-conditioned media on endothelial progenitors were weaker than the one of CHIR which the authors suggest could be due to the potency or concentration of ligands in conditioned media. Indeed, there can be numerous hypothesis made here as there can be numerous secreted factors in those conditioned media.

Therefore, instead of including these data (i.e Figure 3) in this manuscript, I would rather suggest the authors to use it as the basis of a study on the contribition of wnt ligands in the effects of neural rosette and/or astrocyte-conditioned media-effects on ECs derived from hPSC. They could make use of their hPSC lines with doxycycline-inducible expression of short hairpin RNAs targeting CTNNB1 to confirm that neural rosette and/or astrocyte-conditioned media-effects in ECs (notably on GLUT-1, Claudin 5, caveolin-1) are β-catenin dependent.

In such experiment a comparison of the effects of wnt7A, neural rosette and astrocyte-conditioned media (+ brain pericyte conditioned medium or even coculture) on hPSC-derived endothelial progenitors (including from doxycycline-inducible expression of short hairpin RNAs targeting CTNNB1) would probably clarify the role of wnt ligand in the effects of neural rosette and astrocyte-conditioned media (or even coculture).

– Page 10-Line 217 (and associated Figure 6C): "We first confirmed that the process of dextran internalization required the membrane fluidity of an endocytosis-dependent process by carrying out the assay at 4C”

Although I get the idea of comparing fluorescent dextran internalization with or without CHIR treatment, I miss the point of the necessity of comparing the 4C and 37C. I assume 4C would not only affect membrane fluidity but also would reduce any active processes taking place in the cells as well as reduce thermal molecular agitation within the assay.

Therefore, I would suggest to remove this experiment at 4{degree sign}C which I find confusing and possibly replace it by experiment performed with inhibitors of the caveolin pathway (e.g Filipin, Genistein, Nystatin,.;)

– Page 21 – Line 483:

Although I agree with the statement that Praça et al., reported moderate improvement in barrier function using a combination of VEGF, Wnt3a, and retinoic acid, in my opinion TEER values should be used intra assay (e.g as in the manuscript to compare the effects of CHIR on TEER) not from one study to the next as several parameters could influence TEER (e.g Temperature, buffer capacity, surface of the insert,…).

*Reviewer #4 (Recommendations for the authors):*

The manuscript by Gastfriend et al., describes a novel approach to generate a blood-brain barrier (BBB) in vitro model from human pluripotent stem cells (hPSCs) by inducing BBB characteristics in yet undifferentiated, naïve endothelial precursor cells with Wnt/β-catenin activating agents. Specifically, the authors identified the inhibitor of glycogen synthase kinase 3β (GSK3β) CHIR99021 (CHIR) to consistently induce crucial BBB-relevant genes such as GLUT-1, Cldn5, LEF1, APCDD1 and ZIC3. At the same time GSK3β inhibition lead to reduced expression of PLVAP. Interestingly, the authors also observed, along with the up-regulation of BBB genes, an up-regulation of caveolin-1 (CAV1) that however, did not result in increased transendothelial permeability. Further the authors compared the treatment with CHIR to the stimulation with recombinant Wnt7a, Wnt7b and combinations thereof, as well as to the treatment with conditioned medium (CM) from astrocytes and neuroblast rosettes. Wnt7a and the combination of Wnt7a and Wnt7b showed a pronounced effect on endothelial purity during IPSC differentiation, but considerably lower effects on the expression of barrier genes than CHIR. Neuroblast and astrocyte CM showed effects on endothelial purity and neuroblast CM also some effect on GLUT-1 expression.

The authors also investigated the barrier characteristics at different stages of endothelial progenitor differentiation, i.e. at passage 1 and passage 3, as well as at a higher maturity at passages 4 and 5. The presented data suggest that at passage 1 and 3 CHIR is able to induce expression of GLUT-1, claudin-5, LSR and barrier properties measured by TEER, whereas at later passages 4 and 5 these effects were considerably lower or absent.

Consequently, the author interpret this finding as a a result of endothelial differentiation, suggesting that naïve progenitors are susceptible to Wnt induction, whereas terminally differentiated cells are not or at lower level.

Finally, the authors performed RNA-Seq of CHIR and DMSO treated progenitor cells at passages 1 and 3, as well as comparing Wnt7a and DMSO. Expression profiling demonstrated that CHIR induces crucial BBB genes which essentially overlapped with the inducing effects of Wnt7a. Interestingly, only Wnt7a resulted in the induction of Sox17, which has been shown to be up-regulated in the maturing BBB.

The concept of a time-dependent induction of BBB characteristics by Wnt/β-catenin induction as a “window of opportunity" is interesting and innovative, although others have already proposed this.

The presented data are solid and the manuscript is well written with high standard figures. However, some experimental details and interpretation of the data require the authors attention to augment the scientific merit of the manuscript:

It would be important that the authors clearly claim if the present manuscript describes a novel in vitro model or a basic stem cell biological finding.

The following critical points are related to this first issue because the that

The authors show that the treatment with Wnt7a and in particular with Wnt7b only show little or even no effects on Wnt pathway activation (Figure 8—figure supplement 1 B; no AXIN2 induction by Wnt7a/7b!!) and induction of BBB characteristics. It would be important to further investigate this topic to clearly demonstrate that Wnt7a/7b indeed are functional as recombinant proteins in the described experimental setting. Wnt7 has been shown to be only little diffusible, requiring the direct contact of the sending cell to the receiving cells (Eubelen, M. et al., Science (New York, NY) 361, eaat1178, 2018). Therefore, the authors should elaborate on an experimental setting to stimulate the naïve endothelial progenitors with Wnt7a/7b in a paracrine manner. This would help to clarify the question regarding the discrepancy between the CHIR and the Wnt7a treatment.

Essentially, the same critique can be raised concerning the treatment with conditioned medium from neuroblast rosettes and astrocytes. Although both may express Wnt7a/7b, specifically these Wnts may not accumulate in the CM and hence, the effects may be minor.

Page 16, line 359-363: The assumption made in the sentence “Upon recombination, the Ctnnb1 flex3 allele produces a dominant mutant β-catenin lacking residues that are phosphorylated by GSK-3β to target β-catenin for degradation (Harada et al., 1999); as such, this strategy for ligand- and receptor-independent Wnt activation by β-catenin stabilization is directly analogous to CHIR treatment." might be wrong, as blocking GSK3beta potentially has many other effects than “just" activating β-catenin-mediated transcription!! The authors should discuss this topic in more depth.

One of the highlight of the present manuscript is the inducibility of naïve endothelial progenitors by CHIR compared to differentiated endothelial cells. Hence it would be important to compare the expression pattern by RNA-Seq of these two conditions.

---

## [Author Response]

Essential revisions:1) The authors need to explore in more depth the differences in the expression profile between naïve endothelial progenitors and (partially) differentiated ECs by RNA-Seq.

We agree with the reviewers that a more comprehensive analysis and discussion of transcriptomics data is warranted. We have greatly expanded the analysis of our RNA-seq data, focusing on the attributes of the partially differentiated ECs resulting from CHIR treatment. We have also included new analyses that deploy weighted gene correlation network analyses to distinguish the unique features of CHIR treatment compared with those acquired through extended culture. Finally, we have performed enrichment analyses to compare and contrast CHIR effects to those elicited by direct stabilization of β-catenin. We have substantially revised the corresponding sections of the Results and Discussion to highlight these additional analyses, but given the number of figures that have been edited and added, have not duplicated the revised figures in the response document. In detail:

– We have highlighted additional genes of interest induced by CHIR in Passage 1 ECs, including additional transcription factors also expressed by brain ECs in vivo (*MSX1*, *EBF1*), and *FLVCR2*, a gene with a known role in brain angiogenesis (Figure 7).

From Results: “We also identified upregulated transcription factors: *ZIC3,* which is highly enriched in brain and retinal ECs in vivo and downstream of Frizzled4 signaling (Wang et al., 2012; Sabbagh et al., 2018), and *SOX7*, which acts cooperatively with *SOX17* and *SOX18* in retinal angiogenesis (Zhou et al., 2015), were upregulated by CHIR in our system (Figure 7D). *MSX1* and *EBF1*, which are expressed by murine brain ECs in vivo (Vanlandewijck et al., 2018) were also CHIR-upregulated (Figure 7D). Additional CHIR-upregulated genes included *ABCG2* (encoding the efflux transporter Breast Cancer Resistance Protein, BCRP), *APLN*, a tip cell marker enriched in postnatal day 7 murine brain ECs compared to those of other organs, and subsequently downregulated in adulthood (Sabbagh et al., 2018; Sabbagh and Nathans, 2020), and *FLVCR2*, a disease-associated gene with a recently-identified role in brain angiogenesis (Santander et al., 2020) (Figure 7C-D).”

– We have better enumerated genes induced by CHIR in Passage 3 ECs: In Figure 7—figure supplement 1, we have highlighted genes up- and down-regulated by CHIR in a format analogous to Figure 7; In Figure 7—figure supplement 2, we have identified genes encoding transcription factors, secreted proteins, and transmembrane proteins that are consistently CHIR-upregulated at Passage 1 and Passage 3.

From Results: “In Passage 3 ECs, many of the CHIR-mediated gene expression changes observed at Passage 1 persisted, including *SLC2A1*, *LSR*, *LEF1*, *AXIN2*, *APCDD1*, *ZIC3*, *EBF1*, *FLVCR2* and *ABCG2* upregulation and *PLVAP* downregulation (Figure 7E; Figure 7—figure supplement 1). […] Conversely, *JAM2*, which encodes junctional adhesion molecule 2, a component of EC tight junctions (Aurrand-Lions et al., 2001; Tietz and Engelhardt, 2015), was upregulated by CHIR at Passage 3, but not at Passage 1, as was the retinol-binding protein-encoding gene *RBP1* (Figure 7—figure supplement 1).”

– We used Weighted Gene Correlation Network Analysis to identify highly correlated gene modules with EC passage number and CHIR treatment (Figure 7—figure supplement 3). Importantly, this analysis complements gene-by-gene differential expression analysis by (a) demonstrating that genes central to the module that strongly correlates with CHIR treatment are canonical targets of Wnt/β-catenin signaling, (b) identifying novel genes that strongly correlate with these known targets, and (c) demonstrating that CHIR treatment and passage number have distinct effects on the transcriptome as a whole despite some concordantly regulated genes (e.g., *CLDN5* and *PLVAP*).

From Results: “We used Weighted Gene Correlation Network Analysis (WGCNA) (Zhang and Horvath, 2005; Langfelder and Horvath, 2008) to identify modules containing genes with highly correlated expression across the 14 EC samples (Figure 7—figure supplement 3A; Supplementary file 3). […] Further, despite some similarly-regulated genes between the passage number and CHIR treatment comparisons (e.g., *CLDN5, CAV1*, *PLVAP*), the transcriptional responses to these two experimental variables were globally distinct as assessed by gene correlation network analysis (Figure 7—figure supplement 3B).”

– We have performed enrichment analysis to identify pathways that may be upstream of observed transcriptional changes upon CHIR treatment at passage 1 and 3. We have also performed this analysis on literature data from pituitary ECs to investigate how pathways induced by CHIR-mediated inhibition of GSK-3 might differ from pathways induced by direct stabilization of β-catenin (described in additional detail below under point 2.) These data are in Figure 8—figure supplement 2 and Supplementary file 5.

2) Moreover, the authors should investigate other pathways induced by CHIR (GSK3 substrates; Akt, IκB-α, presenilin etc.) in comparison with dominant-active β-catenin.

We thank the reviewers for the suggestion to compare pathways induced by CHIR (in our system) versus stabilized β-catenin (literature data). For stabilized β-catenin, we focused on the RNA-seq data from the mouse pituitary EC system, as this system demonstrated the strongest transcriptional response to β-catenin stabilization (compared to mouse liver ECs and cultured mouse brain ECs). We tested lists of CHIR- and β-catenin-upregulated genes against the Hallmark gene set collection, a curated database of high-confidence links between genes and potential upstream biological processes and signaling pathways. Complete results are shown in Figure 8—figure supplement 2 and Supplementary file 5, and are summarized in the Results section. Briefly, we found that multiple signaling pathways (including *Wnt/β-catenin signaling*, *Notch signaling*, *TNFα signaling via NF-κB*, *KRAS signaling up*) were enriched in both CHIR-treated and β-catenin-stabilized systems. A key potential difference is enrichment of the *PI3K AKT mTOR signaling* gene set in Passage 1 CHIR-treated ECs; such enrichment is not present in pituitary ECs with β-catenin-stabilization, or Passage 3 CHIR-treated ECs. This result may suggest that CHIR transiently activates the AKT/mTOR pathway. Overall, however, the strong and persistent enrichment of the *Wnt/β-catenin signaling* gene set in CHIR-treated ECs at both Passage 1 and 3 (along with upregulation of canonical transcriptional targets of β-catenin, the centrality of such canonical transcriptional targets in gene correlation network analysis, and the attenuation of GLUT-1 induction with *CTNNB1* knockdown), support the notion that Wnt/β-catenin signaling is a central mediator of our observed phenotypes. We agree that other GSK-3 targets may participate, and believe a direct comparison of CHIR treatment and β-catenin stabilization in the hPSC-derived EC system could be informative in the future. Finally, we note that β-catenin stabilization is also an imperfect mimic of Wnt ligand-induced signaling, as ligand-induced signaling also involves GSK-3 inhibition; this further motivates future work using Wnt ligands and the endogenous Wnt receptors and intracellular transduction machinery.

From Results: “Last, because GSK-3 is a component of numerous signaling pathways in addition to Wnt/β-catenin (Eto et al., 2005; Beurel et al., 2015; Hermida et al., 2017), we used RNA-seq data to infer pathways that might be differentially regulated by the two strategies for activating Wnt/β-catenin signaling employed in the experiments above: CHIR treatment, which increases β-catenin stability by inhibiting GSK-3, or direct stabilization of β-catenin. […] Differences in other aspects of these two experimental paradigms (in vitro versus in vivo, naïve versus CNS-proximal, human versus mouse), however, caution against over-interpretation of these results.”

From Discussion: “CHIR is widely used to activate Wnt/β-catenin signaling in cell culture (Lian et al., 2012, 2014; Patsch et al., 2015; Sakaguchi et al., 2015; Gomez et al., 2019; Pellegrini et al., 2020; Guo et al., 2021). It remains unknown, however, to what extent CHIR-mediated inhibition of GSK-3 in ECs mimics the effects of Wnt ligand-induced inhibition of GSK-3 or direct stabilization of β-catenin. In our system, although the GLUT-1-inductive effect of CHIR was partially inhibited by β-catenin knockdown and our RNA-seq data revealed a transcriptional response characteristic of canonical Wnt signaling, it is possible that CHIR affects other signaling pathways, as suggested by pathway enrichment analysis. Thus, employing ligand-based strategies to activate Wnt signaling will be an important next step. […] Finally, it would also be informative to directly compare CHIR and/or Wnt ligand treatment to direct stabilization of β-catenin in this system, for example, by generating a hPSC line with inducible expression of a dominant active β-catenin.”

3) Given that the mechanistic insight in the discrepancies between the stimulations with CHIR and Wnt7a/b, as well as with conditioned medium from neurospheres and astrocytes is limited, the authors should omit these data and add this approach to the discussion.

As suggested, we have now removed these data from the manuscript and instead added a paragraph summarizing these approaches to the Discussion. While Wnt7a/b and conditioned media treatment are rational and important approaches, we agree that there are many potential explanations for their lack of efficacy (lack of ligand bioactivity, lack of necessary cell-cell contact, lack of ligand accumulation in the medium, lack of necessary receptor machinery on endothelial progenitors). We also agree that further mechanistic examination of this phenomena should be the subject of future work.

From Discussion: “CHIR is widely used to activate Wnt/β-catenin signaling in cell culture (Lian et al., 2012, 2014; Patsch et al., 2015; Sakaguchi et al., 2015; Gomez et al., 2019; Pellegrini et al., 2020; Guo et al., 2021). […] Importantly, neural progenitor cells and astrocytes likely would also contribute other yet-unidentified ligands important for acquisition of CNS EC phenotype.”

4) The authors should clearly delineate the aim of the study that, as perceived by the reviewers, appears to be the characterization of the Wnt pathway effects on directing human pluripotent stem cells toward the differentiation to a BBB endothelium.

We appreciate the suggestion to clarify the aim of the study. We have reworded the thesis statement in the final paragraph of the introduction to better convey the goal of the work:

From Introduction: “In this work, we aimed to define the effects of activating Wnt/β-catenin signaling in hPSC-derived, naïve endothelial progenitors and assess the extent to which this strategy would drive development of a CNS EC-like phenotype.”

Statements in the Abstract and Discussion also emphasize this overarching aim:

From Abstract: “Together, our work defines effects of Wnt activation in naïve ECs and establishes an improved hPSC-based model for interrogation of CNS barriergenesis.”

From Discussion: “In this work, we investigated the role of Wnt/β-catenin signaling on induction of BBB properties in a human EC model, using naïve endothelial progenitors derived from hPSCs.”

Reviewer #2 (Recommendations for the authors):While this paper may be a worthwhile method development paper (as published in PLoS One, Science Reports or Microvascular research; eg. Cecchelli 2014 doi:10.1371/journal.pone.0099733; Linville 2020 doi: 10.1016/j.mvr.2020.104042), it lacks scientific novelty and depth required for a publication in eLife.Numerous papers have identified Wnt3a for mesoderm induction and Wnt7a as the major brain EC ligand, and the use of a chemical GSK-3 inhibitor for more powerful Wnt signaling pathway activation is not novel or surprising. Indeed many other models and Wnt-pathway response papers have been published, numerous in the last two years (e.g. Laksitorini et al., Sci Rep 2019 9(1):19718. "Modulation of Wnt/β-catenin signaling promotes blood-brain barrier phenotype in cultured brain endothelial cells").

We agree that a substantial body of previous research has demonstrated the ability of Wnt signaling to both induce mesoderm differentiation and to induce EC BBB properties, and that many previously reported methods have used GSK-3 inhibition as a Wnt activation strategy. We wish, however, to emphasize the novelty of our work in applying Wnt activation to achieve a BBB-like phenotype in hPSC-derived definitive ECs, and in comprehensively profiling the resulting transcriptome and other aspects of cellular phenotype.

Several previous papers have examined the effects of Wnt activation in immortalized ECs (Laksitorini et al.,) or hematopoietic stem cell-derived ECs (Cecchelli et al.,). In contrast to these cell sources, hPSCs offer the opportunity to track EC differentiation along the entire developmental trajectory, and probe temporal aspects of Wnt activation (discussed further below). We have clarified this point in the Introduction:

From Introduction: “Human pluripotent stem cells (hPSCs) offer an in vitro human model system for systematic investigation of molecular mechanisms of BBB phenotype acquisition, especially given their ability to model early stages of endothelial specification and differentiation.”

The few papers that have attempted Wnt activation in hPSC-derived ECs have not isolated the effects of Wnt and/or performed comprehensive analysis of Wnt-mediated changes. For example, Linville et al., evaluated Wnt7a only in combination with VEGF, and then only on “sprouting” and not on BBB phenotype. Praça et al., demonstrated a slight impact of Wnt3a on permeability, but analyzed protein/gene expression changes with Wnt3a in combination with VEGF or VEGF and retinoic acid. Further, none of the above works have used RNA-sequencing to comprehensively profile the Wnt-regulated EC transcriptome. Other hPSC-derived BBB models for which RNA-seq data are available do not address the impact of Wnt signaling on the transcriptome, do not proceed through definitive endothelial progenitors, and have an underlying epithelial gene signature (Vatine et al., 2017, *Cell Stem Cell*; Qian et al., 2017, *Sci Adv*).

To the best of our knowledge, our work represents the first report that activation of Wnt signaling in hPSC-derived ECs can partially achieve BBB phenotype. We further suggest that our transcriptome data defining Wnt-regulated genes, and importantly, BBB-associated genes not regulated by Wnt, will be an important addition to the literature for those studying human BBB development and those wishing to continue to improve our ability to recapitulate the BBB in the dish.

In addition one major claim of archievement is the analysis of EC maturity linked to Wnt-responsiveness. However, it has already been shown that there is a crucial timing difference in the responsiveness to Wnt signals both in vivo and in vitro (including, but not at all limited to the following examples: Hübner et al., Nat commun 2018 Nov 19;9(1):4860 "Wnt/β-catenin signaling regulates VE-cadherin-mediated anastomosis of brain capillaries by counteracting S1pr1 signaling"; Corada et al., Circ Res 2019 124(4):511-525 "Fine-Tuning of Sox17 and Canonical Wnt Coordinates the Permeability Properties of the Blood-Brain Barrier").Additionally many of the mentioned examples not only analyze the contribution of WNT signaling, but also link other signaling pathways, whereas this paper identifies other but unknown signaling contributions for establishing ECs with full BBB properties.

We thank the reviewer for this comment and we have now expanded the Discussion to incorporate these concepts and citations. We note that these reports have primarily characterized the endogenous temporal profile of Wnt/β-catenin signaling in the context of CNS angiogenesis and barriergenesis, and used loss of function approaches (e.g., dominant negative TCF, IWR-1 treatment) to identify timepoints during normal CNS vascular development at which Wnt/β-catenin is necessary for specific phenotypes. A major conclusion is that the level of Wnt/β-catenin in CNS ECs peaks early in development and subsequently declines, but this result does not directly address the *competence* of ECs to respond to Wnt signals, which varies across different contexts, including time (embryo versus adult), organ (CNS versus CNS-proximal versus non-CNS), and in vitro versus in vivo (e.g., Munji et al., 2019, *Nat Neurosci*; Wang et al., 2019, *eLife*; Sabbagh et al., 2020, *eLife*). To address this gap in understanding of the role of Wnt/β-catenin in imparting BBB properties to ECs, we performed gain of function experiments by activating Wnt/β-catenin with CHIR at different timepoints during the hPSC differentiation to ECs, and found that early ECs were more responsive than those after extended culture. As discussed above, hPSCs are particularly well suited to this type of longitudinal analysis; we further contend that this treatment paradigm is novel and that the results will be a useful contribution to the literature.

From Discussion: “Thus, our results support this hypothesis and suggest that the loss of BBB developmental plasticity in ECs is an intrinsic, temporally-controlled process rather than a result of the peripheral organ environment. The molecular mechanisms underlying this loss of plasticity remain poorly understood. While previous studies have demonstrated that the level of Wnt/β-catenin signaling in CNS ECs peaks early in development and subsequently declines (Corada et al., 2018; Hübner et al., 2018), this finding does not address mechanisms underlying the competence of ECs (CNS and non-CNS) to respond to Wnt signals. In RNA-seq data of Passage 3 control (DMSO-treated) ECs, *LEF1* and *TCF7* were strongly downregulated compared to Passage 1 cells. This result suggests that low baseline expression of these transcription factors, which form a complex with nuclear β-catenin to regulate Wnt target genes, may partially explain the poor efficacy of CHIR in matured ECs, although additional work is necessary to assess functional relevance of these differences.”

Given the lack of novelty and the following mentioned difficulties of biological sample variations, I do not recommend the paper for publication in eLife.

Please see the above description of novelty and impact.

Other comments:Figure 1 shows more than factor2 differences between different differentiation samples (in terms of glut-1 positive cells), making this a less robust model than indicated.

We agree that the percentage of GLUT-1^+^ ECs resulting from Wnt3a treatment appears to differ between the two differentiations performed; in each case, however, Wnt3a significantly increased the percentage of GLUT-1^+^ ECs over control (Differentiation 1, control: 1.3±0.4%, Wnt3a: 9.1±1.0%, *P* = 0.00028, Student’s *t* test; Differentiation 2, control: 0.5±0.3%, Wnt3a: 2.9±0.4%, *P* = 0.00096, Student’s *t* test). We feel it is important to present both differentiations to provide a realistic picture of expected variability, which, unlike with immortalized cell lines, is an inherent aspect of hPSC differentiation. We also believe that this variability, as well as the small proportion of GLUT-1^+^ ECs elicited by Wnt3a treatment, motivates our subsequent use of CHIR.

The overall percentage of Glut-1 positive cells in Figure 1 and 2 is very low, making the comparison between the effectiveness of the different ligands while statistically significant compared to the control, most likely not significant compared between the ligands. Therefore the claims of e.g. Wnt7a being more effective is based on one of 3 samples of the same differentiation. All these comaparative claims will have to be either supported by much more data or taken out. The only clear result is that CHIR can stimulate expression of Glut-1 in around 90% of the cells, whereas the offered Wnt ligands affect 2-4%. To me this suggests that soluble WNT ligands in the culture medium are least effective in stimulating a response. It would have been most interesting to explore if co-cultures and hence the supply in e.g. Cytonems or other vesicle or membrane-bound delivery systems would make a huge difference. However this is not the scope of the paper.The analysis of mean fluorescence intensity of claudin-5, caveolin-1, and GLUT-1 normalized to Hoechst mean fluorescence intensity within the area of claudin-5+ ECs only is less informative, as one highly expressing cell will outweigh many low expressing cells and there is no published link of an endothelial cell resembling a BBB endothelial cells more closely in correlation with their Glut-1 expression intensity.

As discussed under “Essential Revisions” point 3, we have removed data on Wnt7a/b from the manuscript and instead focus now in Figure 2 on the effect of CHIR. Because CHIR induces GLUT-1 expression in ~90% of ECs, and images do not indicate substantial heterogeneity in claudin-5 or caveolin-1 immunofluorescence across ECs, we believe mean fluorescence intensity within EC colonies is a reasonable metric for comparison of the DMSO and CHIR conditions. We have added replicates from several additional differentiations to Figure 2 to accurately convey robustness of these effects. Moreover, the protein level increases are also quantified in Figure 4 and are concordant with those determined by quantitative image analysis.

Unfortunately, when analysing the effects of conditioned media in Figure3 only those relative quantifications were used and again the spread between the two different differentiation rounds was so high, that a comparative analysis is hindered and not represented by statistical analysis compared to control only. For example the Caveolin/Hoest values of the orange dataset were increased in the CHIR treatment, but unchanged in Control and NR-CM, but in contrast in the green dataset the NR-CM resulted in a higher index compared to CHIR or the control. Again were no statistical comparison was done between samples, the drawn conclusions can not indicate there are any.

We agree that the discrepancy between two differentiations in the conditioned medium-elicited changes in caveolin-1 (and claudin-5) fluorescence intensity motivates additional replication of these treatments. As discussed under “Essential Revisions” point 3, we have removed the conditioned medium data from this manuscript and will instead address this in future work.

Figure 5 Anal;yses the effect of passage number in combination with CHIR treatment. However the individual differentiation samples again behave partially opposite: the green sample Glut1/ß-actin is only weakle elevated in passage 1 and upregulated after passage3, the purple sample is high in passage 1 and becomes downregulated after passage 3.

The data shown in panels B and D are not from matched differentiations. We recognize that the color scheme may have implied this, and we have adjusted the colors accordingly. Thus, quantitative comparison of data in panels B and D is not possible. We have also added data from one additional differentiation to Figure 4D to better estimate the expected GLUT-1 induction at Passage 3.

Figure 7 the effects in the mature EC should be directly compared to the effects in more naïve from the same differentiation.

We thank the reviewer for suggesting this key experiment, which we have now performed and the results of which we have included as a revised Figure 6 (formerly Figure 7). In a direct comparison of GLUT-1 fluorescence intensity in ECs from the same differentiation, CHIR treatment initiated at the early timepoint (D5; “Passage 0”) led to a ~70-fold increase, while no statistically significant increase was observed when treatment was carried out at Passage 4. Similarly, CHIR-mediated increases in EC number and claudin-5 abundance were observed when treatment was carried out at D5, but not when treatment was carried out at Passage 4. These results strengthen the conclusion that EC responsiveness to Wnt activation decreases with extended culture.

From Results: “To test this hypothesis, we matured hPSC-derived ECs in vitro for 4 passages (until approximately day 30) prior to initiating CHIR treatment for 6 days, and compared the resulting cells to differentiation-matched samples treated with CHIR immediately after MACS (Figure 6A). Both Passage 1 DMSO-treated ECs and Passage 5 DMSO-treated ECs, which are analogous to EECM-BMEC-like cells we previously reported (Nishihara et al., 2020), did not have detectable GLUT-1 expression (Figure 6B). Compared to DMSO controls, the CHIR-treated Passage 5 ECs exhibited no increase in GLUT-1 abundance (Figure 6B-D), which contrasts with the marked increase observed when CHIR treatment was initiated immediately after MACS (Figure 6B-D). Furthermore, CHIR treatment in matured ECs did not increase claudin-5 expression and did not increase EC number (Figure 6B-D), in contrast to the increases observed in both properties when treatment was initiated immediately after MACS (Figure 6B-D).”

Reviewer #3 (Recommendations for the authors):Although I have a positive opinion on the manuscript, I personally think that the manuscript could be slightly simplified (and shorten). Therefore, I have listed below several points that I think should be addressed (and possibly used to shorten the manuscript) by the authors before publication of the manuscript.– Page 8 Line 173- 192 : "In the CNS, neural progenitors ….. ….. the Wnt ligand signal (analyzed further below)”.This part and associated figure 3 compare the effects of neural rosette-conditioned medium and astrocyte-conditioned medium to the one of WNT7A and CHIR based on the fact that both cell types were suggested to express WNT7A (Vatine et al., 2016; Shang et al., 2018)..Although the data are interesting per se., these data in my opinion, do not really have an added value to the main topic of the study (i.e deciphering the effects of wnt signaling in human endothelial cell progenitor with a focus on barrier formation).In addition, the reported effects of neural rosette and astrocyte-conditioned media on endothelial progenitors were weaker than the one of CHIR which the authors suggest could be due to the potency or concentration of ligands in conditioned media. Indeed, there can be numerous hypothesis made here as there can be numerous secreted factors in those conditioned media.Therefore, instead of including these data (i.e Figure 3) in this manuscript, I would rather suggest the authors to use it as the basis of a study on the contribition of wnt ligands in the effects of neural rosette and/or astrocyte-conditioned media-effects on ECs derived from hPSC. They could make use of their hPSC lines with doxycycline-inducible expression of short hairpin RNAs targeting CTNNB1 to confirm that neural rosette and/or astrocyte-conditioned media-effects in ECs (notably on GLUT-1, Claudin 5, caveolin-1) are β-catenin dependent.In such experiment a comparison of the effects of wnt7A, neural rosette and astrocyte-conditioned media (+ brain pericyte conditioned medium or even coculture) on hPSC-derived endothelial progenitors (including from doxycycline-inducible expression of short hairpin RNAs targeting CTNNB1) would probably clarify the role of wnt ligand in the effects of neural rosette and astrocyte-conditioned media (or even coculture).

We thank the reviewer for the suggestion to simplify the manuscript and focus here on the effects of CHIR. We agree that the data on Wnt7 and conditioned media treatments would best serve as the basis for a future study where their ability to activate Wnt/β-catenin signaling, and the β-catenin dependence of observed responses, could be unambiguously confirmed. As discussed above under “Essential Revisions” point 3, we have therefore removed these data from the present manuscript, focus on the effects of CHIR in the Results, and mention future avenues for research involving Wnt ligands and conditioned media in the Discussion.

– Page 10-Line 217 (and associated Figure 6C): "We first confirmed that the process of dextran internalization required the membrane fluidity of an endocytosis-dependent process by carrying out the assay at 4C”Although I get the idea of comparing fluorescent dextran internalization with or without CHIR treatment, I miss the point of the necessity of comparing the 4C and 37C. I assume 4C would not only affect membrane fluidity but also would reduce any active processes taking place in the cells as well as reduce thermal molecular agitation within the assay.Therefore, I would suggest to remove this experiment at 4{degree sign}C which I find confusing and possibly replace it by experiment performed with inhibitors of the caveolin pathway (e.g Filipin, Genistein, Nystatin,.;)

While 4°C indeed affects membrane fluidity and thereby reduces/eliminates endocytosis processes, we better clarified the processes by which dextran is internalized by these cells as requested by the reviewer. We repeated the internalization assay in the presence of nystatin (inhibitor of caveolin-mediated endocytosis), chlorpromazine (inhibitor of clathrin-mediated endocytosis), or rottlerin (inhibitor of macropinocytosis). We also used confocal microscopy to visualize dextran and identify whether dextran colocalized with caveolin-1^+^ puncta. In general, these assays suggest that clathrin-mediated endocytosis and macropinocytosis are the predominant pathways of dextran internalization under the conditions of this assay, although a small number of dextran^+^ caveolin-1^+^ puncta were apparent. We have included these data in Figure 5—figure supplement 1, and have also moved the 4°C control to Figure 5—figure supplement 1 to simplify the main figure.

From Results: “We confirmed that the dextran signal measured by this assay was endocytosis-dependent by carrying out the assay at 4°C and with inhibitors of specific endocytic pathways (Figure 5—figure supplement 1A-C). Compared to vehicle control, chlorpromazine (inhibitor of clathrin-mediated endocytosis) and rottlerin (inhibitor of macropinocytosis) both decreased dextran uptake, while nystatin (inhibitor of caveolin-mediated endocytosis) did not significantly affect uptake (Figure 5—figure supplement 1B,C), consistent with the very small number of dextran+ caveolin-1+ puncta observed by confocal imaging (Figure 5—figure supplement 1D).”

– Page 21 – Line 483:Although I agree with the statement that Praça et al., reported moderate improvement in barrier function using a combination of VEGF, Wnt3a, and retinoic acid, in my opinion TEER values should be used intra assay (e.g as in the manuscript to compare the effects of CHIR on TEER) not from one study to the next as several parameters could influence TEER (e.g Temperature, buffer capacity, surface of the insert,…).

We included the TEER value of ~60 Ω×cm^2^ reported by Praça et al., to draw attention to the similarity in order of magnitude to our results, and agree completely that TEER values obtained in two different assays/laboratories are not directly comparable. We have therefore reworded this sentence to clarify this point:

From Discussion: “For example, Praça *et al.,* showed that a combination of VEGF, Wnt3a, and retinoic acid directed EPCs to brain capillary-like ECs with moderate transendothelial electrical resistance (TEER) similar in order of magnitude to that reported here.”

Reviewer #4 (Recommendations for the authors):[…]The presented data are solid and the manuscript is well written with high standard figures. However, some experimental details and interpretation of the data require the authors attention to augment the scientific merit of the manuscript:It would be important that the authors clearly claim if the present manuscript describes a novel in vitro model or a basic stem cell biological finding.

Our primary goal was to provide insight into the cellular phenotype elicited by activation of Wnt/β-catenin signaling in naïve hPSC-derived endothelial cells, particularly as it relates to gain of CNS EC phenotype. As discussed above under “Essential Revisions” point 4, we have clarified this goal in the Introduction. We do not believe the two goals described by the reviewer are mutually exclusive, however, and we would suggest that others consider this hPSC-based definitive EC in vitro model system for future studies of CNS EC barriergenesis.

From Introduction: “In this work, we aimed to define the effects of activating Wnt/β-catenin signaling in hPSC-derived, naïve endothelial progenitors and assess the extent to which this strategy would drive development of a CNS EC-like phenotype.”

The following critical points are related to this first issue because the thatThe authors show that the treatment with Wnt7a and in particular with Wnt7b only show little or even no effects on Wnt pathway activation (Figure 8—figure supplement 1 B; no AXIN2 induction by Wnt7a/7b!!) and induction of BBB characteristics. It would be important to further investigate this topic to clearly demonstrate that Wnt7a/7b indeed are functional as recombinant proteins in the described experimental setting. Wnt7 has been shown to be only little diffusible, requiring the direct contact of the sending cell to the receiving cells (Eubelen, M. et al., Science (New York, NY) 361, eaat1178, 2018). Therefore, the authors should elaborate on an experimental setting to stimulate the naïve endothelial progenitors with Wnt7a/7b in a paracrine manner. This would help to clarify the question regarding the discrepancy between the CHIR and the Wnt7a treatment.Essentially, the same critique can be raised concerning the treatment with conditioned medium from neuroblast rosettes and astrocytes. Although both may express Wnt7a/7b, specifically these Wnts may not accumulate in the CM and hence, the effects may be minor.

We agree that the lack of upregulation of canonical transcriptional targets (e.g., *AXIN2*, *LEF1*) in the RNA-seq data of Wnt7a/b-treated ECs is strong evidence that these ligands are not effective in activating Wnt/β-catenin signaling in this system. As described under “Essential Revisions” point 3, we have thus removed these data from the paper. In the Discussion, we have also added suggestions of coculture-based strategies that may mitigate the poor solubility/diffusability of Wnt ligands:

From Discussion: “Given evidence that Wnt ligands have poor solubility (Janda et al., 2012) and our preliminary data suggesting that supplementation of culture medium with Wnt7a and Wnt7b is largely ineffective in activating Wnt/β-catenin signaling in this system, special emphasis should be placed on strategies that present Wnt7a, Wnt7b, and/or Norrin in a manner that concentrates ligands at the cell surface, for example, by using direct cocultures of endogenously Wnt-producing cells (neural progenitors or astrocytes) or Wnt-overexpressing cells.”

Page 16, line 359-363: The assumption made in the sentence “Upon recombination, the Ctnnb1 flex3 allele produces a dominant mutant β-catenin lacking residues that are phosphorylated by GSK-3β to target β-catenin for degradation (Harada et al., 1999); as such, this strategy for ligand- and receptor-independent Wnt activation by β-catenin stabilization is directly analogous to CHIR treatment." might be wrong, as blocking GSK3beta potentially has many other effects than “just" activating β-catenin-mediated transcription!! The authors should discuss this topic in more depth.

We have reworded the sentence to clarify this important difference between the two approaches, and added additional data analysis and discussion of this topic to both the Results and Discussion sections (as discussed above under “Essential Revisions” point 2).

From Results: “this strategy for ligand- and receptor-independent Wnt activation by β-catenin stabilization is similar to CHIR treatment, although GSK-3 phosphorylates targets other than β-catenin (discussed below).”

One of the highlight of the present manuscript is the inducibility of naïve endothelial progenitors by CHIR compared to differentiated endothelial cells. Hence it would be important to compare the expression pattern by RNA-Seq of these two conditions.

We agree that comparing the endothelial progenitors (day 5 cells) to matured ECs (Passage 4) would provide valuable insight into to their differential responsiveness to CHIR. Although we do not have RNA-seq data perfectly matched to these two time points, we can use our RNA-seq data of Passage 1 and Passage 3 control (DMSO-treated) ECs as a surrogate for this comparison. We found marked downregulation of *LEF1* and *TCF7* at Passage 3, and suggest this as a possible explanation for the attenuated CHIR-responsiveness of matured ECs. We have added these data to Figure 7—figure supplement 4 and to the Results section. We also added this observation to the Discussion section and clearly highlighted the imperfect nature of the comparison and the need for additional follow-up studies:

From Results: “We also evaluated transcript-level expression of components of the Wnt signaling pathway in Passage 3 control (DMSO-treated) ECs as a first step towards understanding the relative lack of responsiveness observed when CHIR treatment was initiated in matured (Passage 4) ECs (Figure 7—figure supplement 4). While *CTNNB1*, *GSK3B*, and genes encoding components of the destruction complex were not significantly different between Passage 3 and Passage 1, *LEF1* and *TCF7* were strongly downregulated in Passage 3 cells (Figure 7—figure supplement 4B).”

From Discussion: “In RNA-seq data of Passage 3 control (DMSO-treated) ECs, *LEF1* and *TCF7* were strongly downregulated compared to Passage 1 cells. This result suggests that low baseline expression of these transcription factors, which form a complex with nuclear β-catenin to regulate Wnt target genes, may partially explain the poor efficacy of CHIR in matured ECs, although additional work is necessary to assess the functional relevance of these differences.”